# Cyclooxygenase-2 (COX-2) as a Target of Anticancer Agents: A Review of Novel Synthesized Scaffolds Having Anticancer and COX-2 Inhibitory Potentialities

**DOI:** 10.3390/ph15121471

**Published:** 2022-11-26

**Authors:** Noor ul Amin Mohsin, Sana Aslam, Matloob Ahmad, Muhammad Irfan, Sami A. Al-Hussain, Magdi E. A. Zaki

**Affiliations:** 1Department of Pharmaceutical Chemistry, Faculty of Pharmaceutical Sciences, Government College University, Faisalabad 38000, Pakistan; 2Department of Chemistry, Government College Women University, Faisalabad 38000, Pakistan; 3Department of Chemistry, Government College University, Faisalabad 38000, Pakistan; 4Department of Pharmaceutics, Faculty of Pharmaceutical Sciences, Government College University, Faisalabad 38000, Pakistan; 5Department of Chemistry, Faculty of Science, Imam Mohammad Ibn Saud Islamic University (IMSIU), Riyadh 11623, Saudi Arabia

**Keywords:** drug design, hybrid molecules, cancer cell lines, molecular docking, COX-2

## Abstract

Cancer is a serious threat to human beings and is the second-largest cause of death all over the globe. Chemotherapy is one of the most common treatments for cancer; however, drug resistance and severe adverse effects are major problems associated with anticancer therapy. New compounds with multi-target inhibitory properties are targeted to surmount these challenges. Cyclooxygenase-2 (COX-2) is overexpressed in cancers of the pancreas, breast, colorectal, stomach, and lung carcinoma. Therefore, COX-2 is considered a significant target for the synthesis of new anticancer agents. This review discusses the biological activity of recently prepared dual anticancer and COX-2 inhibitory agents. The most important intermolecular interactions with the COX-2 enzyme have also been presented. Analysis of these agents in the active area of the COX-2 enzyme could guide the introduction of new lead compounds with extreme selectivity and minor side effects.

## 1. Introduction

Cancer is caused by the uncontrolled proliferation of cells and is the most lethal ailment among non-infectious diseases. Invasive cancer is the primary cause of death in developed countries and the second-largest cause of death in developing countries. In 2020, 19.3 million cases and 10 million fatalities were reported due to cancer [1]. In 2022, 1.9 million new cases and 0.6 million deaths are predicted in the United States of America as a result of cancer [2]. The high mortality rate has been maintained over recent decades, despite the increased availability of various anticancer drugs [3]. In the future, cancer incidence is expected to increase, with an estimated 26 million cases predicted in 2030 [4,5]. Still, there is a considerable difference between the supply and requirement of new anticancer agents. The currently available anticancer agents are linked to severe post-treatment adverse effects, due to their lack of selectivity. Therefore, the invention and development of novel and selective anticancer therapies with the fewest side effects and multiple modes of action are urgently needed [6]. The idea that inflammation causes cancer was advanced by Rudolf Virchow after observing the presence of leukocytes in tumor cells [7]. Now, it is a recognized fact that chronic inflammation results in an increased risk of cancer [8]. Various growth factors, such as epidermal growth factor (EGF) and fibroblast growth factors (FG), are released during the process of inflammation, which increases cell proliferation and, hence, causes cancer [9]. Cyclooxygenase (COX) is the fundamental enzyme in the biosynthesis of prostaglandins and is important for the pathophysiological process of inflammation. The COX enzyme occurs in three isoforms, i.e., COX-1, COX-2, and COX-3. COX-1 is found in normal tissues, where it controls prostaglandin production and, hence, is necessary for normal physiological function. COX-2 is produced during the process of inflammation and carcinogenesis [10]. Non-steroidal anti-inflammatory drugs (NSAIDs) obstruct COX-1 and COX-2 enzymes to different extents and reduce prostaglandin biosynthesis [11]. The blockage of COX-1 causes several complications, the most common of which is GIT ulceration [12]. COX-2 is the key inflammatory mediator and its expression is up-regulated during inflammation [13,14,15]. Therefore, COX-2 has been the target of anti-inflammatory agents for many years [16]. The COX-2 enzyme has a vital role in the initiation of cancer because it inhibits apoptosis and starts the process of angiogenesis. An increased COX-2 level has been observed in colorectal (60%), breast (40%), pancreatic, esophageal, and lung cancer, and in melanoma [17,18,19]. NSAIDs (**1**–**6** Figure 1) are frequently used for the management of cancer-related pain and inflammation. The inhibition of COX-2 by some NSAIDs has led to the inhibition of the process of cancer and the induction of metastasis [20]. Therefore, COX-2 has emerged as a possible target for new anticancer drugs and there is increased research in this direction [21]. The selective COX-2 inhibitor, celecoxib (**5**), has established anticancer properties regarding some cancers, such as ovarian cancer and adenomas [22]. COX-2 inhibitors have also shown the capability to overcome multidrug resistance because they reduce the expression of efflux pumps and, hence, reveal a different domain [23]. There is currently a great deal of research into the discovery and synthesis of new COX-2 inhibitors due to their unique role in cancer chemotherapy. This review deals with the new anticancer derivatives synthesized over the last five years that have also exhibited COX-2 inhibitory activity.

## 2. Drug Molecules with Dual Anticancer and COX-2 Inhibitory Activity

Several drug molecules possessing dual anticancer and COX-2 inhibitory properties have already been reported. Licofelone (**7**) (Figure 2) is a dual COX and lipoxygenase (LOX) inhibitor. Licofelone showed anticancer activity in the colon (HCA-7) and breast (MCF-7) cancer cell lines by inducing apoptosis [24]. Compound **8** is a triazole derivative that exhibited anticancer activity and also demonstrated COX-2 and LOX enzyme inhibition. Compound **8** also facilitated the process of apoptosis in the A549 cancer cell line. Another derivative of this class will be discussed later in the triazole group [25]. Similarly, compound **9** is an indole derivative that displayed anticancer and anti-inflammatory activities by inhibiting the COX-2 enzyme [26]. Darbufelone (**10**) also showed anticancer activity in the colorectal cancer (CRC) cell line, accompanied by dual LOX/COX-2 inhibition [27]. Similarly, compound **11** also exhibited dual anticancer and COX-2 inhibitory activities [28]. In the following sections, we have discussed various classes of organic compounds and their anticancer and COX-2 inhibitory activities. 

### 2.1. Pyrazole Derivatives 

Organic compounds exhibiting pyrazole scaffolds have drawn major attention as anticancer agents due to their anti-inflammatory and anticancer activities [29,30,31]. The diaryl heterocycle is a common scaffold in many COX-2 inhibitors because of its greater selectivity for this enzyme and better safety profile. The presence of the aminosulfonyl or sulfonylmethyl group is also very significant for COX-2 selectivity [32,33]. These groups interact with His90, Gln192, and Arg513 in the COX-2 enzyme through hydrogen bonding [34]. Inceller and co-workers designed and synthesized 1,3-diarylpyrazole derivatives. The aromatic rings, such as phenyl and pyridyl, were attached to the 1, 3-diarylpyrazole ring through the oxopropenyl linker group. Upon their evaluation as anticancer agents, **12, 13,** and **14** (Figure 3) displayed excellent anticancer activities (IC_50_ < 10 µM) against breast (MCF-7), liver (Huh-7), and colon (HCT-116) cancer cell lines. Derivatives having a 4-pyridyl ring displayed superior activity compared with the 3-pyridyl derivatives. The introduction of 4-sulfonylmethyl phenyl at position 1 of pyrazole produced inactive compounds. Derivative **14** also displayed excellent antiplatelet (71.43%) and COX-2 (25.45%) inhibitory activities but was not found to be selective for COX-2 because COX-1 inhibitions were also observed [35]. These derivatives showed excellent drug-likeness values and followed the Lipinski rule of five [36]. Mukarram et al. synthesized new 1,3-diaryl pyrazole derivatives, into which the α-difluoro-β-hydroxyl carbonyl fragment was incorporated. Derivatives **15**, **16**, and **17** demonstrated prominent antiproliferative activity against the MCF-7 cell line (40.59–51.03% inhibition), compared to the standard drug methotrexate (54.29–68.42% inhibition) and showed less cytotoxicity for the normal cell line. Derivative **17** is an acid derivative with a terminal carboxyl group, while **15** and **16** are hydrazide derivatives, with a terminal hydrazide group. The hydrazide and acid derivatives presented better activity compared to the ester derivatives. Derivative **17** also showed excellent COX-2 inhibitory (38.56%) activity, while using aspirin (11.11 ± 0.13%) as the standard drug. Compounds having halogen-substituted aromatic rings at position 3 of the pyrazole scaffold presented greater activity levels. Derivative **16** showed prominent DPPH radical scavenging activity (72.76%) when using ascorbic acid (88.63%) as a standard drug [37].

Ren et al. synthesized pyrazole sulfonamide derivatives by incorporating ferrocene in the pyrazole. The nitric oxide (NO) donor molecule was attached to the pyrazole ring. Compound **18** (Figure 4) showed prominent anticancer activity against cervical (Hela; IC_50_ = 0.34 ± 0.22 µM), MCF-7 (IC_50_ = 2.12 ± 0.31 µM), lung (A549; IC_50_ = 2.6 ± 0.44 µM), and murine melanoma (B16-F10; IC_50_ = 3.86 ± 0.55 µM) cancer cell lines by using celecoxib (IC_50_ = 7.55 ± 0.66–25.87 ± 2.01 µM) as the standard drug. Derivative **18** also presented prominent COX-2 (IC_50_ = 0.28 µM: SI = 142.96) inhibitory activity, compared to the standard drug, celecoxib (IC_50_ = 0.38 ± 0.08 µM: SI = 97.29). An evaluation of the mode of action studies showed that **18** induced the process of apoptosis in the Hela cell line in a dose- (1.25–10 µM) and time (0–36 h)-dependent manner [38]. Yamali et al. synthesized new pyrazole derivatives in which positions 3 and 5 were modified. The ester and hydrazide groups were introduced at position 3 and different aryl, as well as heteroaryl, rings were introduced at position 5. Compounds **19** (CC_50_ = 37.7 µM) and **20** (CC_50_ = 58.1 µM) showed less activity against the human squamous cell carcinoma (HSC-2) cancer cell line, compared to the standard drug, doxorubicin (CC_50_ = 0.2 µM). These compounds were found to be more active than fluorouracil (CC_50_ > 400 µM) and methotrexate (CC_50_ > 1000 µM). Derivatives **19** and **20** showed selectivity for the cancer cell lines, and **20** also exhibited the most prominent inhibition of human carbonic anhydrase (hCA-II) isoform-II, with a ki value of less than 7.4 nM. Compounds **19** and **20** are ester derivatives; the conversion of ester into the hydrazide linkage leads to a further decline in activity [39]. Zhang et al. reported new pyrazole derivatives, to which an aminophosphonyl moiety was attached. Upon an evaluation of the anticancer activity, **21** demonstrated excellent anticancer activity (IC_50_ = 4.37 ± 0.49 µM) against the MCF-7 cell line by using paclitaxel (IC_50_ = 5.74 ± 0.39 µM) as a standard drug. Compound **21** also showed inhibition (IC_50_ = 0.22 ± 0.04 µM; SI = 179.18) of COX-2 and presented more selectivity for this enzyme than the standard drug, celecoxib (IC_50_ = 0.39 ± 0.08 µM, SI= 97.79). The substitution of the N-1 aromatic ring at the para position was important for COX-2 inhibition. The fluorine-substituted aromatic rings manifested greater activity, and the replacement of the fluorine atom by the bromine or sulfonamide group produced less active derivatives. Derivatives with a naphthalene ring at position number 5 showed prominent anticancer activity. Compound **21** also showed the process of apoptosis in MCF-7 cells, via mitochondrial depolarization [40].

### 2.2. Pyrazoline Derivatives 

Pyrazoline is a reduced form of pyrazole and has achieved great utility in terms of new drug synthesis. Pyrazoline derivatives have revealed anti-inflammatory, analgesic, anticancer, and antibacterial activities [41,42,43]. Qiu and co-workers investigated the anticancer potential of new pyrazoline derivatives. The pyrazoline ring was attached to aromatic and heterocyclic scaffolds. The sulfonamide group was incorporated at the para position of one aromatic ring. The compounds were evaluated by using molecular docking and the active molecules were then synthesized. Derivative **22** (IC_50_ = 0.08 ± 0.03 µM, SI = 451) (Figure 5) expressed COX-2 inhibition and also demonstrated prominent activity (IC_50_ = 1.63 ± 0.97 µM) against the A549 cancer cell line. Compound **22** also demonstrated the phenomenon of apoptosis in a dose-dependent manner, i.e., 1–10 µM. The derivatives in this series showed low cytotoxicity (CC_50_ > 100 µM) to the normal cell line, 293T. The inhibitory effect of these derivatives was linked to the level of COX-2 in the cells. In docking studies of **22** with COX-2, five hydrogen bonds were observed between the sulfonamide group and the amino acids Gln178, Leu338, Ser339, Arg499, and Phe504. Van der Waal and carbon–hydrogen interactions were also observed [44]. Yan and co-workers synthesized diaryl and heteroaryl dihydropyrazole derivatives in which the sulfonamide group and benzodioxole ring were introduced. Upon evaluation of anticancer activity via an MTT assay, **23** demonstrated excellent anticancer activity (IC_50_ = 0.86 ± 0.02 µM) against the intestinal cancer (SW-620) cell line, compared to the standard drug, celecoxib (IC_50_ = 1.29 ± 0.04 µM). Compound **23** also showed prominent activity against A549 (IC_50_ = 1.94 ± 0.06 µM), Hela (IC_50_ = 2.98 ± 0.17 µM), MCF-7 (IC_50_ = 2.99 ± 0.13 µM), and liver (HepG2; IC_50_ = 2.96 ± 0.14 µM) cancer cell lines. Derivatives having a 1,3-benzodioxole skeleton presented superior activity, compared to 1,4-benzodioxole-derived compounds. These compounds showed less cytotoxicity for the normal cell line. Derivative **23** also exhibited excellent COX-2 blockage (IC_50_ = 0.35 ± 0.02 µM, SI = 137.3), in comparison to the standard drug, celecoxib (IC_50_ = 0.41 ± 0.03 µM, SI = 145.8). Compound **23** also induced apoptosis in a dose-dependent manner (2–8 µM). Compound **23** presented a docking score of −45.96 kcal/mol and interacted, via hydrogen bonding, with the His75, Phe504, and Val102 amino acids of COX-2, in addition to normal interactions [45].

### 2.3. Pyrazole- and Pyrazolone-Based Hybrid Molecules

Molecular hybridization is a trend in drug design and discovery in which the pharmacophores of active drug molecules are combined to produce new molecules. The molecular hybridization approach has also been employed to produce new compounds with dual biological activities [46,47]. In 2019, Abdellatif and co-workers carried out the synthesis of hybrid molecules, based on the activities of pyrazole and thiohydantoin as COX-2 and topoisomerase inhibitors. The methylene group of the thiohydantion nucleus was connected to the carbonyl group of pyrazole molecules. These hybrid molecules showed better inhibition of COX-2, compared to COX-1. The 4-methoxy-substituted agents showed greater activity, compared to the unsubstituted derivatives. Derivatives **24**, **25**, and **26** (Figure 6) demonstrated prominent anticancer potential against MCF-7, A549, and HCT-116 (IC_50_ = 2.78 µM–7.30 µM) cell lines when compared to doxorubicin (IC_50_ = 0.2 ± 0.005–2.44 ± 0.36 µM). Molecules **24**–**26** also showed the prominent inhibition of COX-2 (IC_50_ = 0.65 µM–1.72 µM) and less ulcerogenicity (UI = 3.87) as seen in celecoxib (UI = 2.99). Derivative **26** showed hydrogen bonding with the COX-2 enzyme, involving Lys68 and Tyr108, and presented a −20.4 kcal/mol docking score. Compound **26** also inhibited (IC_50_ = 23.3 µM) the topoisomerase type-2 (Top-2) enzyme by using camptothecin (IC_50_ = 20.2 µM) as the standard [48]. Akhtar and co-workers reported the synthesis of pyrazoline-pyrazole hybrid molecules. The Vilsmeier–Heck reaction was used to prepare the pyrazole aldehyde molecule, which was converted to pyrazoline through the Claisen–Schmidt reaction and then converted into the final products. Derivative **27** appeared to be the most active against A549 (IC_50_ = 4.94 µM), cervical (SiHa; IC_50_ = 4.54 µM), COLO-205 (IC_50_ = 4.86 µM), and HepG2 (IC_50_ = 2.09 µM) cancer cell lines by using 5-fluorouracil (IC_50_ = 2.08 ± 0.01- 19.01 ± 0.11 µM) as the standard. Derivative **27** also showed more inhibition against COX-2 (IC_50_ = 1.09 ± 0.001 µM; SI = 80.03) compared to celecoxib (IC_50_ = 0.26 ± 0.002 µM; SI = 95.84). These derivatives presented less cytotoxicity (IC_50_ = 50 µM) for normal cell-line human keratinocytes (HaCaT). The introduction of an electronegative atom in ring **C** decreased the activity of these derivatives. Compound **27** presented a docking score of −10.66 kcal/mol and interacted via hydrophobic interactions, hydrogen bonding, and π-cationic interactions. The hydrogen atom of the amide group presented hydrogen bonding with Ser353. Arg120 and Arg513 displayed π-cation interactions, while Phe381 and Phe385 showed hydrophobic interactions (π-alkyl) with the methoxy group [49]. Belal and co-workers synthesized benzoxazole/benzothiazole and pyrazole hybrid molecules. Derivative **28** displayed prominent activity against the A549 (IC_50_ = 2.4 µM and SI = 83.2) cancer cell line. Derivative **29** showed moderate activity against MCF-7 (IC_50_ = 34.39 µM, SI = 16.59) cancer cell line. Derivatives **28** (IC_50_ = 0.56 µM, SI = 6.32) and **29** (IC_50_ = 0.74 µM, SI = 7.22) also presented more prominent inhibition of the COX-2 enzyme than the standard drug, celecoxib (IC_50_ = 1.11 µM; SI = 6.61). The anticancer activity of these hybrids was related to COX-2 inhibition. Therefore, the introduction of the pyrazolone ring to benzoxazole or benzothiazole increased the activity, while 4-chlorobenzladehyde-substituted agents showed better activity [50]. 

El-Zahar and co-workers synthesized new hybrid compounds by the attachment of pyrazole/thiazole heterocyclic rings to the triazole scaffold. The thiazole, pyrazole, and thiosemicarbazone derivatives showed significant inhibition of COX-2. Compound **30** (IC_50_ = 0.04 µM; SI= 310 Figure 7) emerged as the most potent inhibitor and showed excellent selectivity for COX-2, followed by **31** (IC_50_ = 1.47 µM; SI = 258) and **32** (IC_50_ = 1.98 µM; SI = 218) by using celecoxib as a standard drug (IC_50_ = 0.094 µM; SI = 294). Upon the evaluation of anticancer activity via a sulforhodamine B (SRB) assay, **30** (IC_50_ = 28.5 µM against A549)**, 31** (IC_50_ = 26.5 µM against HepG2), **32** (IC_50_ = 27 µM against A-549) showed mild anticancer activity (59.05–92.42% inhibition) by using fluorouracil (IC_50_ = 0.52–0.69 µM) as a standard agent. Compound **30** showed the process of apoptosis and demonstrated the excellent inhibition of human carbonic anhydrase (hCA Xii isoform), with a ki value equal to 13.4 nM. The molecular interaction studies of **30** with COX-2 presented hydrogen bonding between sulfonamide oxygen and Arg120, as well as Tyr355. Other hydrogen bonds were observed between Gly354 and the thiazole ring nitrogen, hydrazine NH, and Gln192 [51]. Fadaly and co-workers synthesized new 1,2,4-triazole and pyrazole hybrid molecules, in which the nitric oxide (NO)-releasing group and sulfamoyl groups were introduced. In that study, the authors tried to boost the anticancer and anti-inflammatory activities. Elsewhere, the NO group augmented the impact of anticancer agents by enhancing their invasion into cancer cells and reducing the resistance [52]. Derivatives **33** (IC_50_ = 3.66 ± 0.12–5.34 ± 0.91 µM), **34** (IC_50_ = 4.37 ± 0.27–6.48 ± 0.21 µM), **35** (IC_50_ = 1.48 ± 0.08 µM) and **36** (IC_50_ = 0.33 ± 0.06–6.38 ± 1.19 µM) presented excellent activities against A549, MCF-7, HCT-116, and PC-3 cancer cell lines compared to doxorubicin as standard drug. The sulfamoyl-substituted derivatives showed greater activity compared to sulfonylmethyl and un-substituted analogs, probably due to their resemblance to the celecoxib pharmacophore. These hybrid molecules were found to be more potent COX-2 inhibitors, compared to COX-1. These compounds also showed a substantial release of NO and a lower ulcer index than ibuprofen as the standard. Compounds **33** (IC_50_ = 0.55 µM, SI = 9.78) and **36** (IC_50_ = 0.89 µM, SI = 10.47) appeared to be potent inhibitors of COX-2 while using celecoxib as a standard drug (IC_50_ = 0.83 µM, SI = 8.68). These derivatives interacted with the Ser516 and Tyr317 of COX-2 via hydrogen bonding. Derivatives **35** and **36** exhibited apoptosis and cell-cycle arrest in the PC-3 cancer cell line [53].

Kulkarni and co-workers synthesized coumarin–pyrazolone hybrid molecules and these molecules presented moderate anticancer activities. Compound **38** (Figure 8) showed selectivity against HT-29 (growth inhibition (GI) = 37.13%), lung (NCI-H23; GI = 69.69%), CNS (SNB-75; GI = 34.19%), PC-3 (GI = 30.46%), and breast (T-47D; GI = 44.14%) cancer cell lines. Derivatives **37** (82%)**, 38** (81%), and **39** (76%) also displayed prominent anti-inflammatory activities in an in vitro evaluation via a protein denaturation assay. Both activities are influenced by the presence of the methoxy group at position **6** of the coumarin nucleus; benzene substitution, as well as the phenyl ring at the pyrazolone nitrogen, are important for both activities. The carbonyl group of **37** coumarin showed two hydrogen bonds with Arg121 and Tyr356 of the COX-2 enzyme. The oxygen of coumarin and the amino group of pyrazolone displayed hydrogen bonding with Tyr356 and Ser531. The oxygen of pyrazolone exhibited hydrogen bonding with Arg121 [54]. Li and co-workers synthesized 1,5-diaryl pyrazole and morpholine hybrid molecules and evaluated their anticancer and COX inhibitory activities. Compounds **40** (IC_50_ = 8.16–10.21 µM) and **41** (IC_50_ = 6.43–10.97 µM) presented prominent cytotoxic activities against the MCF-7, A549, Hela, F-10, and 293-T cancer cell lines. These derivatives carry 4-trifluoromethyl phenyl at position 5 of the pyrazole scaffold. The substitution of trifluoromethyl by the methyl, methoxy, or ethoxy groups decreased the activity. Therefore, the introduction of the morpholine ring significantly increased the anticancer activity. Compounds **40** (IC_50_ = 0.19 µM; SI = 162.05) and **41** (IC_50_ = 0.17 µM; SI = 188.58) also demonstrated excellent inhibition levels of COX-2, as well as good selectivity indices, using celecoxib (IC_50_ = 0.25 ± 0.03 µM; SI = 97.88) as standard. Derivative **41** also showed the prominent blockage of LOX (IC_50_ = 0.68 µM) and exhibited the process of apoptosis in a dose-dependent manner by inducing cell-cycle damage in the G2 phase. Compound **41** presented hydrogen bonding interactions with the Arg106, Tyr371, and Ser339 of COX-2. Van der Waals and hydrophobic interactions were also observed [55].

### 2.4. Benzimidazole/Benzoxazole-Based Hybrid Molecules

Abdelgawad and co-workers carried out the synthesis of hybrid molecules, in which phenolic compounds, such as gallic acid, caffeic acid, and coumaric acid, were linked to benzimidazole and the benzoxazole scaffold. Compounds **42** (IC_50_ = 0.9 µM) and **43** (IC_50_ = 0.5 µM) emerged as active inhibitors of EFGR. Derivatives **42** and **43** (Figure 9) showed excellent anticancer activity against lung cancer (H-460 IC_50_ = 1.7 µM) and pancreatic ductal cancer (Panc-1 IC_50_ = 2.8 µM) cell lines, in comparison to erlotinib (IC_50_ = 0.04 ± 0.02–0.05 ± 0.02 µM). In compounds **42** and **43**, gallic acid and caffeic acid are attached to the benzoxazole and benzimidazole rings, respectively. Derivatives **42** (IC_50_ = 4.34 ± 1.87 µM, SI = 3.80) and **43** (IC_50_ = 2.47 ± 1.97 µM, SI = 3.99) also exhibited excellent inhibition of the COX-2 enzyme by using indomethacin as standard (IC_50_ = 3.29 ± 0.5 µM, SI = 0.08). Docking studies of **43** with COX-2 demonstrated hydrogen bonding and arene-cation interactions (Arg513). Hydrogen bonding interactions were observed between the His90, Tyr355, and Ser530 of the COX-2 enzyme and the nitrogen of oxazole, the carbonyl group, and hydroxyl groups [56]. Earlier in 2017, Abdelgawad and co-workers synthesized benzoxazole, benzimidazole, and benzothiazole derivatives, in which a pyrazole ring was incorporated to produce new molecules. Derivative **44** exhibited prominent activity against MCF-7 (IC_50_ = 6.42 ± 1.32 µM) and A549 (IC_50_ = 8.46 ± 1.89 µM) cancer cell lines by using doxorubicin (IC_50_ = 2.11 ± 0.04 µM and 2.74 ± 0.05 µM) as standard. Benzimidazole-based hybrid molecules were found to be more active compared to the benzoxazole and benzothiazole derivatives. Derivative **44** (IC_50_ = 0.12 µM; SI = 104.67) also exhibited the prominent inhibition of COX-2 by using celecoxib (IC_50_ = 1.11 µM; SI = 13.33) as standard. In molecular docking investigations with COX-2, **44** exhibited hydrogen bonding (docking score = −13.22 kcal/mol) interactions with His90, Arg513, and Tyr356, with a distance of 2.33–3.11 Å [57]. In 2019, Abdelgawad and co-workers synthesized hybrid molecules in which benzimidazole/benzoxazole rings were attached to the pyrimidine ring through phenyl diazo linkage. Derivatives **45** and **46** showed excellent anticancer activity against the MCF-7, A549, PC-3, PaCa-2, and HT-29 (IC_50_ = 4.3–8.8 µM) cancer cell lines. However, these derivatives showed more of an inhibitory effect on the COX-1 enzymes (IC_50_ = 1.92–2.76 µM) compared to the COX-2 (IC_50_ = 7.47–8.21 µM) enzymes. Derivative **45** showed three hydrogen bond interactions (docking score = −11.50 kcal/mol) with Arg513, Val523, and Tyr355 of the COX-2 enzyme. The analog **46** manifested better radical scavenging potential compared to compound **45 [58]**.

### 2.5. Natural Product-Based Hybrid Molecules

Natural products have endured thus far and also have an optimistic future in terms of being the origin of active drug molecules [59]. Organic and medicinal chemists are drawing benefits from natural products despite the advent of new drug discovery approaches, such as high-throughput screening and combinatorial chemistry. The chemical scaffolds of natural products are used as lead compounds, and the derivatization of these products produces new compounds [60,61]. Ren and co-workers prepared hybrid molecules in which chrysin was attached to 1,5-diarylpyrazole derivatives. Chrysin was selected due to its wide range of activities. Derivative **47** (Figure 10) demonstrated prominent anticancer activity against the Hela (IC_50_ = 1.12 ± 0.62 µM) and MCF-7 (IC_50_ = 3.23 ± 0.55 µM) cancer cell lines and manifested less toxicity for the normal cell line, 293T (IC_50_ = 193.94 ± 2.85 µM). Derivative **47** also showed COX-2 (IC_50_ = 2.23 µM, SI = 206.45) inhibitory activity by using celecoxib (IC_50_ = 0.36 ± 0.05 µM; SI = 130.25) as standard. The introduction of electron-donating groups on the phenyl rings of pyrazole showed better results than the electron-withdrawing groups. Derivative **47** showed the phenomenon of apoptosis and achieved cell-cycle arrest in the G1 phase. Compound **47** interacted with COX-2 (docking = −83.77 kcal/mol) and the oxygen atoms of chrysin formed two hydrogen bonds with the Arg222 and His207 amino acids. Van der Waals interactions were also observed [62]. Shen and co-workers synthesized new hybrid molecules by combining pyrazole and coumarin to produce more potent and less toxic agents. The two pharmacophores were joined by using different linker groups to increase the polarity and flexibility. Upon its evaluation as an anticancer agent, **48** showed prominent activity against the A549 (IC_50_ = 4.48 ± 0.57 µM) and Hela (IC_50_ = 5.51 ± 1.28 µM) cancer cell lines, in comparison to the standard drug, celecoxib (IC_50_ = 7.68 ± 0.55–11.06 ± 0.93 µM). Compound **48** carries the trimethoxy phenyl group at position 5 of pyrazole, which displayed extra hydrogen bonding with the target proteins. The longer linker group between pyrazole and coumarin may be the reason for its increased potency. Derivative **48** showed the process of apoptosis and blocked the cell cycle in the G2 phase. Compound **48** also showed the prominent inhibition of COX-2 (IC_50_ = 0.23 ± 0.16 µM; SI = 230) and LOX (IC_50_ = 0.87 ± 0.07 µM) enzymes, in comparison to celecoxib (IC_50_ = 0.41 ± 0.28 µM; SI = 88). Docking studies of **48** with the COX-2 enzyme showed that it interacted (score = −64.387 kcal/mol) with Trp387, His386, Tyr385, and Thr212. Therefore, the introduction of the coumarin ring to the pyrazole scaffold enhanced the anticancer activity of the resulting derivatives [63]. El-Miligy and co-workers synthesized thymol-4-thiazolidinone hybrid molecules, in which the pharmacophoric features of the COX-2 inhibitor, 5-LOX, and PIM-1 (proviral integration Moloney) kinase, were combined. Compound **49** showed the process of apoptosis in the Caco-2 (66.38 ± 0.83%) and HCT-116 (65.27 ± 0.38%) cancer cell lines. Compound **49** (72.33 ± 4.54%) also showed anti-inflammatory activity compared to diclofenac (52.53 ± 4.59%) and also appeared to be the most potent inhibitor (IC_50_ = 0.091 ± 0.0016 µM; SI= 103) of the COX-2 enzyme, compared to standard celecoxib (IC_50_ = 0.045 ± 0.79 µM, SI = 327). Compound **49** carries the ethanoic acid group at position 5 of the thiazolidinone ring. Compound **49** also showed the inhibition of PIM-1 (IC_50_ = 2.96 ± 0.12 µM) and PIM-2 (IC_50_ = 1.38 ± 0.05 µM) kinases, compared to the standard drug staurosporine (IC_50_ = 1.34 ± 0.15 µM; IC_50_ = 0.662 ± 0.02 µM). Compound **49** expressed (docking score = −8.09 kcal/mol) hydrophobic interactions (Val355, Leu517, Trp373, Met508, and Val509) as well as hydrogen bonding (Arg499, His75, Tyr371, Gln178, and Gly512) with COX-2 [64].

### 2.6. Derivatives of Natural Compounds

Marine-derived natural products show excellent biological activities; the structural modification of marine-derived compounds can enhance the pharmacological activities of the lead compound [65,66,67]. Chen et al. synthesized sclerotiorin derivatives in which the pyran ring was modified to produce amine derivatives. The amino group was attached to various acyclic, alicyclic, and heterocyclic scaffolds. Derivatives **50**–**54** (Figure 11) showed prominent anticancer activity versus the A549 cancer cell line, presenting IC_50_ values in the range of 6.39–9.08 µM. Compounds **51**, **53**, and **54** also showed prominent COX-2 inhibitory activities, ranging from 46.7% to 58.7%. Therefore, an association was observed between COX-2 inhibitory activity and cytotoxicity. Derivative **55** presented excellent inhibition of COX-2 (66.1%) and showed no cytotoxicity. Compound **55** should be further investigated as an anti-inflammatory agent. The docking interaction of **50** with COX-2 revealed that the ester linkage interacted with Gln92, while the hydrophilic groups showed hydrogen bonding with COX-2. The alkyl chains and cyclohexene showed hydrophobic interactions [68]. El-Naggar et al. synthesized new ricinin derivatives and evaluated their activity against oral cell carcinoma. Compound **56** showed excellent cytotoxicity (69.22% inhibition; IC_50_ = 90 µM) in comparison to the standard drug, fluorouracil (100%). Compound **56** possesses a biphenyl group that is linked to the ricinin scaffold. Derivative **56** showed docking scores of −7.8 kcal/mol and −7.6 kcal/mol with protein tyrosine phosphatase (PTP1B) and COX-2, respectively. In terms of interactions with PTP1B, hydrophobic interaction was observed with Phe182. Derivative **57** showed the highest docking score of −8.5 with the COX-2 enzyme. The carbonyl group of compound **56** is hydrogen-bonded with the Tyr385 of COX-2. These derivatives were found to be non-toxic for normal cells. Therefore, it was concluded that compound **56** presented anticancer activity by the inhibition of COX-2 and PTP1B [69].

Musa and co-workers synthesized new chalcone derivatives as COX-2 and EGFR inhibitors based on the anti-cancer and anti-inflammatory properties of chalcones. Derivatives **58** (IC_50_ = 0.9 µM) and **59** (IC_50_ = 0.8 µM) (Figure 12) exhibited excellent activity against the PaCa-2 cancer cell line. The presence of a 4-fluoro substituent resulted in the formation of potent anticancer compounds. The activity was related to the α, β-unsaturated ketone, fluoro and carboxyl groups. Derivatives **58** (IC_50_ = 0.8 µM) and **59** (IC_50_ = 1.1 µM) showed kinase inhibition against EGFR, in comparison to erlotinib (IC_50_ = 0.05 ± 0.02 µM). Compounds **58** (IC_50_ = 1.27 µM) and **59** (IC_50_ = 1.88 µM) showed excellent inhibition of COX-2 enzyme, compared to indomethacin (IC_50_ = 3.28 ± 0.5 µM). Derivative **59** also showed the prominent inhibition of IL-6 (77%) and TNFα (76%). Compound **59** displayed hydrogen bonding with Asn34 and Arg44, along with sigma-pi/alkyl-pi interaction with Leu152, Lys468, and Pro153 of COX-2. Halogen bonding was seen with Asn43 [70].

### 2.7. Quinoline Derivatives

Manohar and co-workers synthesized new quinoline acetohydrazide and hydrazone derivatives as anticancer agents while using a drug repurposing strategy [71]. Derivatives **60** (IC_50_ = 2.416 µM), **61** (IC_50_ = 2.071 µM) and **62** (IC_50_ = 2.224 µM) exhibited excellent anticancer activity against the MCF-7 cancer cell line, compared to doxorubicin (IC_50_ = 2.444 µM). Derivatives **61** (IC_50_ = 0.13 µM) and **62** (IC_50_ = 0.14 µM) presented the prominent inhibition of the COX-2 enzyme, compared to the standard drug (IC_50_ = 0.38 µM). Derivatives with a trifluoromethyl substituent showed excellent activities. Molecular docking studies revealed that the amino (NH) group, methoxy (OCH_3_), and the nitrogen atom of quinoline interact with the COX-2 enzyme through hydrogen bonding [72]. Compounds **61** and **62** (Figure 13) also followed the Lipinski rule of five, except for some violations [36]. Pallavi et al. synthesized quinoline glycoconjugates, based on the activity of quinoline and its glycoconjugates. Compound **63** produced a maximum cytotoxic effect (53% inhibition) against the Hela cancer cell line. Derivative **63** also followed the pattern of concentration-dependent killing. Compounds **63** (80%) and **64** (83%) showed prominent COX-2 inhibitory activities. A molecular docking investigation with 4DEP presented docking scores of −6.1 and −5.6 kcal/mol. The carbonyl group of quinoline carboxylic acid and the carbonyl group of sugar molecules exhibited intermolecular interactions. In the case of **63**, the oxygen of the amide group exhibited hydrogen bond interactions with Trp50 and Lys74. The NH group of amides interacted with the Trp50 of interleukin 1β (IL-1β) [73].

### 2.8. Quinazolinone Derivatives

Sakr and co-workers synthesized new quinazolinone analogs, in which quinazolinone was attached to ibuprofen, indole acetamide and thioacetohydrazide scaffolds (Figure 14). Compounds **65** (IC_50_ = 15.42 ± 0.06 µM), and **66** (IC_50_ = 13.42 ± 0.17 µM) showed acceptable anticancer activities versus the HT-29 cell line. Compounds **65** (IC_50_ = 0.04 ± 0.08 µM), **66** (IC_50_ = 0.07 ± 0.22 µM) and **67** (IC_50_ = 0.037 ± 0.20 µM) also demonstrated prominent COX-2 inhibitory activities compared to standard celecoxib (IC_50_ = 0.04 ± 0.20 µM). Analogs **66** (49.47%) and **67** (45.37%) also displayed noticeable in vivo anti-inflammatory potential that was comparable to ibuprofen (47.18%) and celecoxib. Derivative **65** has an indole ring in its structure and the aromatic ring is attached to a chlorine atom at the para position. Compound **65** formed a hydrogen bond with Ala527, while compound **67** formed two hydrogen bonds with the Arg120 and Val523 of COX-2. These molecules have acceptable physicochemical properties, which are within the limits of the Lipinski rule of five [74]. El-Sayed and co-workers synthesized new Schiff bases of quinoxaline and quinazoline-4-one with aromatic and heteroaromatic aldehydes. Compound **68** demonstrated anticancer activity toward HCT-116 and the LoVo cell line, presenting IC_50_ values of 217 µM and 277 µM, respectively. Compound **68** was found to be non-toxic for the normal cell line. Derivatives **68**–**70** showed prominent suppression (> 80%) of the COX-2 enzyme. Derivative **68** showed hydrogen bonding with the Tyr355 and Arg120 of COX-2 through the oxygen and nitrogen of the aminoquinazoline scaffold. Furthermore, π-π interactions were observed with Tyr355 through the methoxyphenyl ring. The azomethine (CH = N) linkage was involved in hydrogen bonding with Arg120. The methoxy group of quinazolinone presented hydrophobic interactions with Met522. Compounds with a methoxy group and a chlorine atom attached to the quinazolinone core showed prominent activity. Structure **68** showed favorable pharmacokinetic properties, such as absorption, distribution, metabolism, and excretion, which makes it the leading compound for future modifications [75].

### 2.9. Indole- and Indanone-Based Molecules

Sever et al. synthesized new indomethacin-based triazolo [3,4-b]-1,3,4-thiadiazine derivatives, based on the known activity of thiadiazine derivatives. Compound **71** (Figure 15) showed anticancer activity toward the glioma cell line (T98) and also showed apoptosis by 11% and 12% at doses of 50 µM and 100 µM. Structure **71** also showed an interaction with the COX-2 enzyme in a similar way to indomethacin. The increased activity of **71** was correlated to the introduction of the 4-methyl group, which has +π and −δ effects. In the molecular docking interactions, **71** showed a similar binding interaction with COX-2 as with indomethacin, involving the amino acids His90, Arg120, Leu352, Tyr355, and Tyr385. Structure **71** also decreased the COX-2 mRNA level [76]. Kumari and co-workers synthesized the c-glycosides of indole by using isoxazole and pyrazoline as linker groups. The carbohydrate molecule was introduced to incorporate stereochemical diversity in these molecules. Upon evaluation of anticancer activity, analogs **72** (IC_50_ = 0.71 µM) and **73** (IC_50_ = 0.67 µM) expressed excellent anticancer activities against the MCF-7 cell line as compared to the standard drug, YM155. These derivatives were found to be non-toxic for the normal cell line, MCF-10. Compound **73** showed prominent inhibition of the COX-2 enzyme (61%) and there was an association with anticancer activity (IC_50_ = 0.67 µM). Analogs with isoxazole as a linker group demonstrated prominent activities. Molecular docking investigations showed that carbohydrate fragments are linked with the COX-2 enzyme via hydrogen bonding [77]. Belgin et al. synthesized a new series of triazolothiadiazine in which an indole scaffold was incorporated. Upon the evaluation of anticancer activity by an MTT assay, **74** showed anticancer activity against the A549 and Caco-2 cell lines. Compound **74** also presented apoptosis via mitochondrial depolarization. The cytotoxic effect was related to COX-2 inhibitory activity. In molecular docking investigations, **74** exhibited interactions with the COX-2 active site and interacted with the His90, Arg120, Gln192, and Tyr355 amino acids [78].

Naaz et al. reported new indole-3-glyoxamide hybrid molecules, using 1,2,3-triazole molecules as a linker group. The click chemistry technique was used to produce new compounds. Compounds **75** (IC_50_ = 8.17 µM, Figure 16) and **76** (IC_50_ = 18.53 µM) showed antiproliferative activity toward the prostate cancer cell line (DU-145), compared to the standard drug, VP16 (IC_50_ = 9.8 µM). The substitution of the ethyl group by chlorine, methyl, cyano, or the 2,4-dichloro group produced less active derivatives. Therefore, the introduction of the 4-tolyl sulfonyl group leads to decreased anticancer activity because the NH group is involved in the hydrogen bonding interaction with Val238. Derivatives **75** and **76** demonstrated the prominent suppression of the COX-2 enzyme (IC_50_ = 0.12 µM; SI = 7.72) compared to the standard drug, celecoxib. Structure **75** (76.9% inhibition) also showed prominent in vivo anti-inflammatory activity, compared to indomethacin (66% inhibition). Derivative **75** also showed excellent COX-2 and 5-LOX blockage. In docking interactions with COX-2, **75** interacted with Arg120 and Ser530 by hydrogen bonding. Hydrophobic interactions were observed with Phe381, Tyr355, and Tyr385 [79]. Abolhasani et al. reported the synthesis of indanone, containing spiroisoxazoline, in which a COX-2 inhibitor was attached to the anticancer agent. In these studies, **77** emerged as the most effective blocker of the COX-2 (IC_50_ = 0.07 µM, SI = 173) enzyme. Derivative **77** also showed prominent cytotoxicity against the MCF-7 cell line (IC_50_ = 0.03 ± 0.01 µM), compared to the standard drug, doxorubicin (IC_50_ = 0.09 ± 0.02 µM). Structures with the methoxy group were more active and the substitution of the methoxy group by a chlorine atom decreases the activity of these derivatives. The oxygen of the methoxy group is involved in hydrogen bonding interactions with the Tyr385 of COX-2 and results in greater selectivity. The nitrogen atom of spiroisoxazoline is also involved in hydrogen bonding. These derivatives showed less toxicity for normal cell lines (MCF-10A and HEK-293), with an IC_50_ value greater than 100 µM [80].

### 2.10. Pyrrolizines as Anticancer and COX-2 Inhibitors

Pyrrolizine is a promising scaffold and has displayed both anticancer and anti-inflammatory activities. Pyrrolizine is being considered in the context of new drug design, due to its multiple pharmacological activities [81,82]. In 2017, Gouda and co-workers synthesized pyrrolizines-5-carboxamide in a continuation of the structure–activity relationship studies of licofelone. Into these analogs, thiophene and naphthalene rings were introduced. Derivative **78** (Figure 17) presented prominent activity toward the MCF-7 cell line (IC_50_ = 4.72 µM). This derivative is attached to the para-chloro-substituted aromatic ring. Compound **79** was most active toward the A549 (IC_50_ = 3.24 ± 0.3 µM) and Hep3B (IC_50_ = 8.69 ± 0.4 µM) cancer cell lines. Compounds **79** (IC_50_ = 0.11 µM) and **77** (IC_50_ = 0.36 µM) also achieved better inhibition of COX-2 than COX-1. Compound **79** showed hydrogen bonding with the Tyr355, Arg120, and Phe518 of COX-2. The naphthyl ring showed hydrophobic interactions with Leu352 and Trp387. The 4-tolyl also formed hydrophobic interactions. The pyrrolizine formed π-cationic interactions with Arg513. Compound **79** also showed the inhibition (7–12%) of various kinases, such as BRAF, MSK1, PDK1, and EGFR. A cell cycle and apoptotic assay showed that **79** is accumulated in the *S* phase and also showed the phenomenon of apoptosis [83]. In 2019, Attalah and co-workers carried out further modifications of the pyrrolizine–indolizine scaffold, in which ethyl benzoate esters were introduced. The structural features of licofelone and ketorolac were combined to produce new derivatives. The anticancer activity was evaluated via an MTT assay. Derivatives **80** (IC_50_ = 0.02 ± 0.01 µM), **81** (IC_50_ = 0.15 ± 0.07 µM), and **82** (IC_50_ = 0.37 ± 0.05 µM) presented prominent activity against the MCF-7 line, compared to lapatinib (IC_50_ = 6.80 ± 1.20 µM). The replacement of the pyrrolidine ring with piperazine produced less active anticancer agents. Structure **82** appeared to be the most active anti-inflammatory derivative (46.32% inhibition), compared to ibuprofen (53.78%). Compound **82** also showed the inhibition of COX-2 (IC_50_ = 2.146 ± 0.79 µM; SI = 13.19), compared to the standard agent celecoxib (IC_50_ = 0.579 ± 0.08 µM; SI = 74). Compound **82** showed a binding free energy of −12.26 kcal/mol toward COX-2 and formed three hydrogen bonds with the Phe518, Ile517, and Tyr355 of COX-2. Furthermore, pi-alkyl, pi-cationic, and pi-sulfur interactions were also observed. The derivatives in this series followed the Lipinski rule of five, as well as the Veber rule, Ghose rule, and Egan and Muegge rule for drug-likeness [84,85]. Shawky and co-workers synthesized pyrrolizine 5-carboxamide derivatives with Schiff bases and thiazolidinone scaffolds. The molecules were designed by combining the structural features of licofelone and tyrosinase kinase inhibitors. Compounds **83** and **84** showed prominent cytotoxicity (IC_50_ = 0.10–0.89 µM) toward the HT-29 and MCF-7 cancer cell lines, compared to the standard drug, lapatinib (IC_50_ = 6.80–12.67 µM). Replacement of the alkyl chains at position 6 with aromatic or heteroaromatic rings produced less active derivatives. Structure **84** (IC_50_ = 13.49 ± 0.63 µM, SI > 7) showed some selectivity for COX-2 and presented apoptosis by blocking the cell cycle in the *S* phase. In the case of thiazolidinone analogs, **85** demonstrated potency against the MCF-7 (IC_50_ = 0.16 ± 0.08 µM) cell line, while **86** showed prominent activity for the A2780 (IC_50_ = 0.11 ± 0.01 µM) and HT-29 (IC_50_ = 0.12 ± 0.03 µM) cancer cell lines. Structure **86** (IC_50_ = 1.27 ± 0.066 µM; SI > 80) also displayed the inhibition of COX-2, with excellent selectivity. These derivatives presented hydrogen bonding and different types of hydrophobic interactions with COX-2 [86].

### 2.11. Xanthone, Naphthoquinone, and Benzothiazine Derivatives

Miladiyah et al. synthesized new xanthone derivatives to which hydroxyl and halogen substituents were attached. Derivative **87** (Figure 18) was the most potent (IC_50_ = 37.79 µM; SI = 66.40) against the colorectal cancer (WiDR) cell line, compared to doxorubicin (IC_50_ = 3.04 µM; SI = 49.38). The molecular docking investigation of compound **87** showed that it is involved in interactions with telomerase, COX-2, and cyclin-dependent kinase (CDK). This compound was found to be non-toxic for the normal VERO cell line. Derivative **87** is the trihydroxyxanthone derivative. Compound **87** showed interactions with the Arg120, Tyr355, and Tyr385 of COX-2. An equation was developed from the QSAR calculations, in which the logP and dipole moment values were positive, indicating the significance of these parameters [87]. Kavalisukas and co-workers synthesized 1,4-naphthoquinone derivatives into which aromatic rings were introduced. Compounds **88** (IC_50_ = 5.8 µM) and **89** (IC_50_ = 20.6 µM) showed excellent activity toward the A549 cancer cell line and manifested minimum cytotoxicity for the normal cell line. The attachment of the phenylamino ring at position 3 of naphthoquinone was significant for anticancer activity. The addition of substituents, i.e., F, Cl, Br, and OC_2_H_5_ groups in the phenylamino ring further decreased the activity. Compound **88** also exhibited excellent pharmacokinetic properties and followed the Lipinski rule of five. Compound **88** presented a binding energy of −9.4 kcal/mol and interacted with COX-2 via hydrogen bonding, van der Waal interactions, and π-sulfur interactions [88]. Rai and co-workers evaluated the in vivo anticancer activities of the 1,4-benzothiazine hydrazone derivatives in dimethylhydrazine (DMH)-induced cancer models. Compounds **90** and **91** produced anticancer effects via inhibiting COX-2-induced JAK-2 and STAT-3 phosphorylation. These derivatives showed no toxicity at a 25 mg/kg bodyweight dose. Derivatives **90** and **91** showed protective effects and reduced elevated levels of the hepatic enzymes AST, ALT, LDH, and ALP. These derivatives restored the glutathione concentration in the body and also reduced the level of COX-2 enzymes at a 25 mg/kg bodyweight dose. Overall, these agents demonstrated prominent anticancer activity in inhibiting CRC and are excellent candidates for future investigations [89].

### 2.12. Miscellaneous Fused Heterocyclic Rings 

Condensed heterocyclic rings containing nitrogen or sulfur as heteroatoms are very important and are commonly used in the synthesis of biologically active molecules. Razik and co-workers synthesized pyrazolo[3,4-d]pyrimidine analogs as the bio-isosteres of the purine nucleus, in which the imidazole ring of purines was replaced by the pyrazole ring. The pyrazole ring was added to increase the level of COX-2 inhibitory activity. The pyrazolo[3,4-d]pyrimidine nucleus was attached to various substituted phenyl piperazine rings, through amide linkers. The anticancer activity was evaluated by the WST-1 assay. Derivative **92** (Figure 19) was found to be the most active (IC_50_ = 11.9–23.7 µM) against MCF-7, MDA-MB-231, human astrocytoma (SF-268), and human skin melanoma (B16-F10) cell lines, compared to cisplatin (IC_50_ = 15.2–21.3 µM). Compound **92** (IC_50_ = 2.97 µM; SI = 2.94) also expressed selectiveness for COX-2, in comparison to celecoxib (IC_50_ = 0.8 µM; SI = 8.61), upon evaluation by the inhibition of LPS-induced COX-2 expression [90]. Hawash and co-workers synthesized new benzodioxole acetate derivatives in which different halogen-substituted aromatic rings were introduced. Compounds **93** (CC_50_ = 0.228 µM) and **94** (CC_50_ = 1.61 µM) were found to be the most effective inhibitors of the Hela cell line upon evaluation via an MTS assay. The ester derivatives presented superior activity, compared to the acid derivatives. The ortho-halogenated derivatives proved more active compared to the meta-substituted derivatives and it was assumed by the authors that ortho-substituted derivatives push the aromatic ring out of the plane of the benzodioxole ring. Compounds **93** (IC_50_ = 1.30 µM; 1.12 µM for COX-2 and COX-1) and **94** (IC_50_ = 1.45 µM; 1.13 µM for COX-2 and COX-1) expressed the non-selective inhibition of COX-2 and the COX-1 enzyme, in comparison to ketoprofen (IC_50_ = 0.158 µM, 0.031 µM for COX-2 and COX-1) [91]. Movahed and co-workers synthesized new compounds possessing a pyrazino[1,2-a]benzimidazole scaffold. Compound **95** (74.8% inhibition) displayed the most effective anticancer activity against the MCF-7 cancer cell line, compared to cisplatin (76.2%). Compound **95** (IC_50_ = 0.11 µM; SI = 909) also showed substantial blockage of the COX-2 enzyme. Derivative **96** (IC_50_ = 0.08 µM; SI = 795) also presented prominent activity as a COX-2 inhibitor. The selectivity of these agents for the COX-2 enzyme was due to the larger pocket of COX-2 than COX-1; the presence of a sulfonylmethyl group was vital for the potency of these derivatives. A correlation was found between the anticancer and COX-2 inhibitory activities of these derivatives. Compound **96** is involved in interactions with the His90, Arg120, Tyr355, and Arg513 of COX-2. The capacity of the COX-2 inhibitor to counteract platelet aggregation is regarded as a prophylactic action to reduce the side effects of CVS. Derivatives **95** (80.74%) and **96** (78.02%) also showed the inhibition of platelet aggregation [92]. Kirwen et al. synthesized 2,3-diaryl-substituted imidazo[4,5b]pyridine derivatives. Upon the evaluation of anticancer activity via an MTT assay, these derivatives displayed mild activity toward MCF-7, MDA-MB-231, SaoS-2, and k-562. This prominent activity was displayed against the k-562 cancer cell line, with IC_50_ values ranging from 42–57 µmol/L. Compound **97** showed the prominent obstruction of COX-2 and displayed an IC_50_ value of 9.2 ± 0.02 µmol/L. In the molecular docking investigation of **97** against COX-2, the n-phenyl ring, imidazo[4,5b]pyridine, and the chlorine atom of the phenyl ring interacted with the COX-2 enzyme [93].

### 2.13. Heterocyclic Compounds as Anticancer and COX-2 Inhibitors

Heterocyclic compounds are extensively utilized in drug design and synthesis, owing to their broad range of pharmacological activities [94]. There are many known heterocyclic compounds, and new heterocyclic compounds are also being discovered, due to ongoing research in synthetic chemistry.

#### 2.13.1. Imidazolone and Triazole Hybrid Derivatives

Lamie et al. synthesized new 1,2-diaryl-4-substituted benzylidene-5(4H)-imidazolone analogs. The benzoxazole/benzothiazole was connected to the imidazolone nucleus through the aromatic linker groups (**98**–**102**; Figure 20). The new agents resemble COX-2 inhibitors due to the presence of vicinal diaryl heterocyclics. Analogs **99** (IC_50_ = 12.8 ± 3 µM) and **100** (IC_50_ = 15.2 ± 1.7 µM) showed moderate activity against the hepatocellular carcinoma (Hep-3B) cell line. Compounds **101** (IC_50_ = 0.84 µM; SI = 3.67) and **102** (IC_50_ = 1.5 µM; SI = 3.06) expressed excellent inhibition of the COX-2 enzyme and showed more selectivity than celecoxib (IC_50_ = 0.071 µM; SI = 3.66). Compound **98** (IC_50_ = 0.02 µM) was the most active inhibitor of LOX compared to the standard drug, zileuton (IC_50_ = 0.83 µM). The presence of the methoxy group in aromatic aldehyde showed selectivity for the COX-2 enzyme. In the molecular mode of action, these derivatives interacted mainly with the His90 of COX-2 via the N atom of benzoxazole/benzothiazole [95]. Cai et al. reported hybrid molecules in which caffeic acid was linked with a COX-2 inhibitor diaryl-1,2,4-triazole by using ester and amide linkages. Hybrid molecules with an amide linkage were found to be important COX-2 blockers, while the ester analogs displayed prominent anticancer activity. Compound **103** was found to be the most active anticancer agent against Caco-2, A549, B16-F10, and the prostate (PC-3) cancer cell lines (IC_50_ = 6.78 ± 0.21 µM to 9.05 ± 0.07 µM), compared to the standard drug cis-diaminplatinum dichloride (IC_50_ = 6.93 ± 0.01 µM to 9.71 ± 0.11 µM). Compound **103** displayed better COX-2 (IC_50_ = 0.21 ± 0.05 µM) inhibition than COX-1 (IC_50_ = 20.5 ± 1.52 µM). In **103**, the methylsulfonyl group also showed hydrogen bonding with His89 and Arg513. The trifluoromethyl-substituted phenyl group exhibited hydrophobic interactions with Phe518, Trp387 and Leu352. The attachment of electron-withdrawing groups at the N-1 of triazole resulted in the formation of potent anticancer and COX-2 agents. Compound **103** also showed in vivo anticancer activity (58.9% inhibition) in a mouse model, with BF-16-F10 melanoma at a dose of 40 mg/kg of body weight [25].

#### 2.13.2. Thiazole Derivatives as COX-2 Inhibitors

Abdelazeem and co-workers synthesized a new series of diphenylthiazole derivatives, in which diphenylthiazole was attached to different aryl rings to produce new compounds. Upon the evaluation of anticancer activity, **104** and **105** (Figure 21) were found to be the most effective anticancer agents against MCF-7 (IC_50_ = 8.02 ± 1.31 µM; 6.25 ± 0.84 µM), HT-29 (IC_50_ = 9.12 ± 1.65 µM; 5.21 ± 1.51 µM) and A549 (IC_50_ = 11.63 ± 2.34 µM; 8.32 ± 1.55 µM) cell lines in comparison to doxorubicin (IC_50_ = 0.90 ± 0.59–1.41 ± 0.63 µM). Structure **104** carries a methyl-substituted aromatic ring attached with the thiourea group, while **105** is the Schiff base derivative. Compounds **104** (IC_50_ = 0.12 µM; SI = 6.96) and **105** (IC_50_ = 0.96 µM; SI = 7.60) were also found to be effective inhibitors of COX-2 but showed less selectivity in comparison to celecoxib (IC_50_ = 0.05 µM; SI = 294). Structure **104** was evaluated for its inhibitory effect against three cancer targets: tubulin, EGFR, and BRAF. Compound **104** was found to be an effective inhibitor of EFGR (IC_50_ = 0.4 µM) and BRAF (IC_50_ = 1.3 ± 0.4 µM), compared to erlotinib (IC_50_ = 0.08 ± 0.04 µM and 0.06 ± 0.02 µM). These derivatives showed a strong correlation between COX inhibition and anticancer activity. In a molecular docking investigation with EGFR, **104** and **105** showed hydrogen bonding with Asp831, while diphenylthiazole showed hydrophobic interactions with the Leu694, Pro770, Gly722, and Asp776 amino acids. The p-tolyl group formed π-π interactions, as well as cationic-π interactions with the Phe699 and Lys721 amino acids. The binding mode of these compounds with EGFR was similar to that of the standard drug, erlotinib [96].

#### 2.13.3. Tetrazole Derivatives

El-Barghouthi et al. reported 1,5-disubstituted tetrazole derivatives by the modification of some existing agents. The methylsulfonyl-substituted aromatic ring was introduced at position 5 of the tetrazole ring. Compound **106** displayed prominent activity against the MCF-7 (IC_50_ = 13.52 µM) and K-562 (IC_50_ = 26.28 µM) cancer cell lines, compared to the standard drug, doxorubicin (IC_50_ = 0.35 µM; IC_50_ = 0.62 µM). Compound **106** showed less toxicity for the normal cell line (HDFa; IC_50_ > 100 µM). Compound **106** (IC_50_ = 1.37 µM; SI = 73) also expressed the blockage of COX-2 in comparison to the standard drug, celecoxib (IC_50_ = 0.02 µM; SI = 350). The analog **106** presented hydrogen bond interaction with the Phe518, Arg513, and His90 of COX-2. The ester linkage interacted with Tyr355 and Arg120. The benzyl group showed hydrophobic interactions, with a pocket comprising Leu92 and Val116 [97].

#### 2.13.4. Oxadiazole and Pyrimidine Derivatives

El-Syed et al. synthesized new oxadiazole hybrid derivatives, in which fragments of EGFR and COX-2 inhibitors were combined. Derivatives **107**–**109** (Figure 22) presented excellent cytotoxicity. Compound **108** expressed significant activity toward the EGFR kinase (IC_50_ = 0.275 µM) in comparison to erlotinib (IC_50_ = 0.417 µM). Derivatives **107** (IC_50_ = 0.4132 ± 0.022 µM) and **109** (IC_50_ = 1.128 ± 0.045 µM) also presented excellent EGFR inhibition. Compound **108** also showed excellent inhibition (IC_50_ = 5.75 ± 0.06 nM) of the renal cell line (UO-13), compared to doxorubicin (IC_50_ = 7.45 ± 0.03 µM). Compounds **107** (IC_50_ = 8.60 ± 0.05 µM) and **109** (IC_50_ = 13.56 ± 0.09 nM) also presented cytotoxicities against the UO-13 cell line. Derivative **107** appeared to be the most active COX-2 inhibitor (IC_50_ = 0.041 µM; SI = 89.7), compared to the standard drug, celecoxib (IC_50_ = 0.049 µM; SI = 308.16). The attachment of the pyridine ring to oxadiazole produced potent derivatives. In compound **107**, the 4-nitrophenyl ring is connected to oxadiazole through the methine linkage and substitution of the nitro group by bromine or chlorine atoms, producing less active derivatives. Analogs with short linker groups presented better activity, compared to longer linker groups. In a molecular docking investigation with COX-2, **107** showed some additional interactions with COX-2, involving Ser516 and Val335. An in silico pharmacokinetic evaluation of these derivatives revealed these compounds to be orally active [98]. Akhtar and co-workers synthesized 5-cyanopyridine derivatives as anticancer and anti-inflammatory agents. Structure **110** showed superior activity against ovarian cancer (GI_50_ = 0.33 µM; SI = 4.4) compared to 5-fluorouracil (GI_50_ = 4.433 µM). Structures **110** (IC_50_ = 0.91 µM; SI = 105) and **111** (IC_50_ = 1.24 µM; SI = 85) also showed the selective inhibition of COX-2 compared to the standard drug, celecoxib (IC_50_ = 0.26 ± 0.002 µM; SI = 95.84). The attachment of the cyano group was essential for the activity of these derivatives. The substitution of the fluorine atom by the methoxy group produced less active compounds. Isobutyl-carrying derivatives exhibited better activity than isopentyl derivatives. Compound **110** presented (docking score = -8.34 kcal/mol) polar interactions with His90, Gln192, Ser353, and Ser530 amino acids. Hydrophobic interactions were observed with Tyr355, Leu359, Met113, Val116, Tyr385, Leu384, Ala527, Ile517, Phe518, and Ala516 [99]. Akhtar and co-workers carried out the further modification of cyanopyrimidine derivatives, in which sulfur-containing isobutyl and isopropyl groups were introduced. Compound **112** exhibited the prominent inhibition of the non-small lung cancer (NCI-H460; GI = 81.34%) and renal cancer (ACHN GI = 72.64%) cell lines upon evaluation via an MTT assay. The derivative **113** displayed activity (GI = 78.84%) against the CNS cancer cell line. Derivatives **114** (GI = 157.7%) and **115** (GI = 152.04%) demonstrated activity against the melanoma cell line. Compounds **112** (%age inhibition = 66.54 ± 0.07), **113** (%age inhibition = 62.77 ± 0.74), and **114** (%age inhibition = 70.25 ± 0.24) showed significant in vitro anti-inflammatory activity at a dose of 100 µg/mL. Derivatives **112**–**115** showed prominent inhibition (IC_50_ = 1.12 ± 0.002–1.52 ± 0.005 µM and SI = 62.04–86.48) of the COX-2 enzyme, in comparison to celecoxib (IC_50_ = 0.26 ± 0.002 µM; SI = 95.84). Compound **114** (docking score = −7.26 kcal/mol) interacted with His90, Gln192, Ser353 and Ser530 amino acids of COX-2. Hydrophobic interactions were established with Tyr355, Leu359, Met113, Val116, Tyr385, Leu384, Ala527, Ile517, Phe518 and Ala516. The secondary amine and cyano groups were found to be significant for these derivatives [100].

Omar et al. synthesized new pyrimidine and triazolopyrimidine derivatives as the bioisosteres of purine molecules. Compound **116** (Figure 23) showed outstanding activity against the MCF-7 cancer cell line (IC_50_ = 8.88 ± 1.3 µM) that was comparable to the standard drug, fluorouracil (IC_50_ = 8.93 ± 1.8 µM). Derivative **117** showed anticancer activity against the HepG2 (IC_50_ = 20.5 µM) cancer cell line, in comparison to fluorouracil (IC_50_ = 19.9 µM). These derivatives also showed selectivity (SI= 10.75, 5.33) for these cancer cell lines. Structure **116** also showed selectiveness (IC_50_ = 0.13 µM; SI= 78.46) for the COX-2 enzyme, rather than the standard drug, celecoxib (IC_50_ = 0.049 µM and SI = 308.16). Compound **117** also showed drug-like properties and followed the Lipinski rule of five. Compound **118** showed prominent anticancer activity (IC_50_ = 8.68 ± 0.2 µM; SI= 13.86) against the Caco-2 cancer cell line, in comparison to fluorouracil (IC_50_ = 4.01 ± 1.2 µM). Compound **118** also showed the selective inhibition (IC_50_ = 0.10 µM; SI= 132) of COX-2 and also antioxidant (EC_50_ = 45.23 µg/mL) activity. The sulfur atom of compound **118** showed hydrogen bonding with the Leu338 of COX-2. Hydrophobic connections were observed with Arg499, Val509, and Ala513. The pyrimidine ring of compound **118** showed arene-H interaction with Ser339. These derivatives showed a lack of hydrogen bonding with COX-1 and, hence, were found selective for COX-2 [101]. Ghorab and co-workers synthesized new hybrid molecules in which thiourea was attached to the nitrogenous heterocyclic compounds having a sulfonamide group. Derivative **119** showed excellent activity against the Caco-2 (IC_50_ = 14 ± 0.5 µM) cancer cell line but **120** showed mild activity against the MCF-7 (IC_50_ = 34 µM), PC-3 (IC_50_ = 26 µM) cell lines, compared to cisplatin (IC_50_ = 5 ± 0.1 µM). Some of the derivatives showed a connection between COX-2 inhibition and anticancer activity. It was observed in molecular docking investigation that these derivatives interact through hydrophobic interaction involving thiourea and hydrogen bonding through the sulfonamide group of COX-2. Compound **120** expressed three hydrogen bonds: one with Trp373, and two with Asn386. Structures **119** (IC_50_ = 0.49 ± 0.021 µM; SI = 6.06) and **120** (IC_50_ = 0.49 ± 0.021 µM; SI = 6.06) were found to be the most effective inhibitors of COX-2 [102].

#### 2.13.5. Thiadiazole and Cyclopentole Derivatives

Raj and co-workers evaluated the anticancer activity of 1,3,4-thiadiazole-derived compounds (**121** and **122**, Figure 24) in a dimethylhydrazine (DMH)-induced colorectal cancer model and investigated the mode of action at the molecular level. These compounds corrected the DMH-induced biological parameters, such as body weight, tumor volume, etc., and also restored the hepatic enzymes, such as AST, ALT, ALP, and LDH, in serum. These derivatives also reinstated the elevated levels of COX-2, Il-2, and IL-6 to normal levels. The molecular mode of action studies showed that these compounds inhibited the COX-2- and IL-6-mediated JAK-2 and STAT-3. The in vitro results were also confirmed by the in vivo results and these agents were found to be safe at a dose of 25 mg/kg body weight [103]. El-Husseiny and coworkers synthesized 2-cyclopentole oxyanisole-derivative compounds. Analogs **124** (IC_50_ = 1.08 µM) and **125** (IC_50_ = 1.88 µM) showed prominent COX-2 inhibitory activity compared to the standard drug, celecoxib (IC_50_ = 0.38 µM). Compounds **123** and **125** showed more prominent anticancer activity (IC_50_ = 4.38 ± 0.4 µM–11.45 ± 1.1 µM) versus the MCF-7, HepG2, PC-3, HCT-116, and Hela cancer cell lines than with the standard drug, doxorubicin (IC_50_ = 5.4 ± 0.25–11.4 ± 1.26 µM). Therefore, the replacement of cycloalkanone with piperidone increased the anticancer activity. Similarly, the piperidine derivatives showed greater activity, as compared to the pyridine derivatives. Compounds **123** (IC_50_ = 5.62 µM) and **125** (IC_50_ = 3.98 µM) also inhibited the PDEB enzyme upon in vitro evaluation. The docking interactions of compounds **124** and **125** revealed that the piperidione fragment interacted with Arg513, His90, Leu352, and Gln192. The carbonyl group of piperidione showed hydrogen bond interactions with the hydroxyl group of Tyr355 and, similarly, other hydrogen bonds were also observed. The methylene (CH_2_) groups of piperidone, cyclopentyl, and methoxy groups also showed hydrogen bonding. In the case of **125**, the nitrile group formed hydrogen bonding with Tyr355 and Arg120. The 4-oxopyrimidine ring showed hydrogen bonding interactions with His90 and Arg513. The methoxy group interacted with Tyr348 by means of hydrophobic interactions [104].

### 2.14. Derivatives of NSAIDs as Anticancer and COX-2 Inhibitors

NSAIDs have also received the attention of medicinal chemists in order to produce new anticancer agents [105,106,107]. Selected COX-2 inhibitors, celecoxib and rofecoxib, showed anticancer potential and, therefore, stimulated further research into NSAIDs [108,109]. Drug repositioning is a technique in which old drugs are tested against new targets rather than synthesizing new ones [110]. The chemical scaffold of NSAIDs, such as indomethacin, celecoxib, rofecoxib, naproxen, and ketoprofen have been modified to produce new analogs. El-Azab and co-workers synthesized thiocarboxylic acid derivatives of NSAIDs as anticancer agents. These derivatives were also evaluated for their COX-1/COX-2 and kinase inhibitory activities. Compounds **126** and **127** (Figure 25) showed excellent anticancer activities (IC_50_ = 7.86 ± 0.39 µM-18.71 ± 1.50 µM) against HepG2, MCF-7 and HCT-116, compared to the standard drugs, 5-fluorouracil (IC_50_ = 5.32 ± 0.17–7.91 ± 0.28 µM) and afatinib (IC_50_ = 5.4 ± 0.25 µM-7.1 ± 0.49 µM). Compounds **126** and **127** are the thiophenol and cyclohexane thiol derivatives of indomethacin and ketoprofen, respectively. Derivatives **126** (IC_50_ = 0.66 µM; SI = 75.8) and **127** (IC_50_ = 0.22 µM; SI = 227.3) also showed the prominent inhibition of COX-2 enzymes and presented excellent selectivity indices. Compound **127** showed hydrogen bonding with Tyr355, electrostatic interaction with Met535, and pi-cationic interaction with Arg513 of COX-2. In the kinase inhibitory activity assay, these derivatives presented very little in the way of inhibitory activities against EGFR, HER2, HER4, and cSrc [111]. Coskun and co-workers synthesized new diflunisal derivatives in which the carboxylic group was converted into thiosemicarbazide and 1, 2, 4-triazole-3-thione. Compounds **128** (IC_50_ = 11.7 ± 2.6 µM), **129** (IC_50_ = 43.4 ± 2.72 µM), and **130** (IC_50_ = 6.2 ± 3.1 µM) presented prominent anticancer activities against PC-3, T-47D and HCT-166 cancer cell lines, respectively, compared to cisplatin (IC_50_ = 0.836–1.25 µM). Compounds including triazole presented prominent activities; the methoxy-substituted compounds proved more active against the T-47D cell line. The thiosemicarbazide functional group showed greater activity against the PC-3 cell line. Fluorine-substituted derivatives were found to be most active, while the substitution of fluorine by chlorine or bromine atoms produced less active derivatives. Derivatives **129** (binding score = −10.57 kcal/mol) and **130** (binding score = −9.60 kcal/mol) presented excellent binding affinities with the COX-2 enzyme by means of hydrophobic interactions (Phe518, Trp387, Tyr385, Tyr348, Phe381, Phe205), hydrogen bonding (Ser530, Tyr355, Glu198, His90), and π-sulfur interactions (Met522) [112].

El-Husseiny and co-workers synthesized new non-carboxylic acid derivatives of naproxen, in which the carboxylic group was converted into cyclic imides, triazole, oxadiazole, and cycloalkane scaffolds. Compounds **131**–**134** (Figure 26) showed excellent anticancer activities (IC_50_ = 4.83–14.49 µM) against MCF-7, Hela, MDA-231, and HCT-116 cancer cell lines in comparison to doxorubicin (IC_50_ = 3.18–26.79 µM). These derivatives also exhibited excellent COX-2 inhibition (IC_50_ = 0.40–1.20 µM) as compared to celecoxib (IC_50_= 0.11 µM). Compounds **133** and **134** carry the arylidene ring and **132** is the oxadiazole derivative. The attachment of the 4-hydroxyphenyl group with oxadiazole produced the COX-2 active derivative. Derivative **134** showed selectivity for COX-2 and presented hydrogen bonding interactions with Ile517 and Gln192 through the hydroxyl group. The methoxy group interacts with Ser530 and Tyr385, while the 4-amino-1,2,4-triazole-5-thione forms hydrogen bonds with Tyr355 and Arg513 [113].

El-Sayyed et al. synthesized new naproxen derivatives in which heterocyclic rings, such as pyridine, pyrazole, pyrazoline, and pyrazolopyridazine were attached to the 6-methoxy naphthalene scaffold. Compounds **135** and **136** (Figure 27) showed 64.6% and 82.6% growth inhibitions, respectively, against non-small cell lung cancers (NCI-H522). Compounds **135** and **136** carry the pyrazole ring attached to the naphthalene, through the carbonyl group. Compound **137** was highly selective toward the MCF-7 cancer cell line, showing 60.9% inhibition. Derivative **137** showed an equal inhibition of COX-1 and COX-2, while **138** showed some selectivity for COX-2. Compound **138** is the hybrid of naproxen with pyrazoline. Therefore, the introduction of pyrazole and pyrazoline rings into the naphthalene scaffold produced interesting activities. In molecular docking investigations with COX-2, **137** mainly showed hydrophobic interactions with the side chains of Val102, Leu517, and Val335, while **138** showed hydrophobic interactions with Leu345, Leu517, and Val509 [114]. Mareedy and co-workers synthesized the nimesulide-1, 2, 3-triazole hybrid molecules. The click chemistry technique was used, in which the nitro group was reduced to the amino group, then into azide, and finally into the triazole molecule. Compounds **139** (IC_50_ = 6.44 ± 0.31 µM against DU-145), **140** (IC_50_ = 5.9 ± 0.15 µM against DU-145), and **141** (IC_50_ = 6.6 ± 0.23 µM against A549) showed prominent anticancer activities against different cancer cell lines. PDE4 has emerged as a new target for anticancer drugs and the docking studies of these compounds with PDE4 showed better scores (−87.42–−94.65 kcal/mol) than nimesulide. In the case of **140**, the NH group of methane sulfonamide presented hydrogen bonding with the Asp346 and Ser442 of the PDE4 enzyme. Therefore, it was assumed that the compound’s anticancer activities are due to its interaction with these proteins [115].

Celik and co-workers synthesized a new ionic liquid by reacting caprolactam with salicylic acid (CL-SA). In the structure determination, two hydrogen bonds were observed between salicylic acid and caprolactam. The dipole moment of compound **142** (Figure 28) was found to be 5.012D, showing its ionic environment. The molecules were evaluated for interaction with COX-2 and DNA Top-II enzymes, via molecular docking. The binding energy of **142** was found to be −8.6 kcal/mol with DNA. The carbonyl group of caprolactam and the hydroxyl group of salicylic acid are involved in hydrogen bonding. Similarly, the docking interactions of **142** with COX-2 yield a binding energy of −8.4 kcal/mol and showed hydrogen bonding with His388. Alkyl and π-alkyl interactions were also observed with Ala202, Val295, and Leu391. The binding energy of this compound with the Top-II enzyme was found to be −7.9 kcal/mol. The evaluation of the SA-CL pharmacokinetic properties and toxicities showed drug-like properties [116]. Punganuru et al. synthesized combretastatin and rofecoxib hybrid molecules, in which the olefinic linkage of combretastatin was replaced with rigid and stable structures. Analog **143**, the new hybrid molecule, showed excellent anticancer activity against the HT-29 (IC_50_ = 258 ± 110 nM) and HCT-116 (IC_50_ = 302 ± 116 nM) cancer cell lines. The structure of the new molecule preserved the cis configuration of combretastatin, due to the presence of a heterocyclic furanone ring, and prevented its conversion into the trans form. This conversion was unsuitable in combretastatin because the trans form was inactive. Compound **143** showed a comparable docking score (−7.87 kcal/mol) with COX-2 and the standard drug, rofecoxib (−8.39 kcal/mol). Compound **143** also presented in vivo anticancer activity in mice infected with the HT-29 cancer cell line and showed the reduction of COX-2 levels. The analog **143** also showed the phenomenon of apoptosis and increased the concentration of the apoptotic proteins, BAX and PUMA. In the molecular docking interaction with COX-2, the furan ring interacted with Val349, Ala527, Leu531, and Arg120. The oxygen atom of the furyl ring showed hydrogen bonding with the Arg120 of COX-2. The iodine atom of methoxy phenyl showed electrostatic interaction with the hydroxyl group of Ser530. The second aryl ring is involved in π-π interaction with Tyr355. Compound **143** can be useful in treating cancer [117].

### 2.15. Metal Complexes as Anticancer Agents and COX-2 Inhibitors

Platinum-containing drugs, such as cisplatin and carboplatin, have been used as anticancer agents but produce severe adverse effects [118]. Currently, metals other than platinum are also being investigated as anticancer and COX-2 inhibitor agents [119]. Obermoser and co-workers synthesized new chlorine-substituted acetylsalicylic acid derivatives, in which cobalt carbonyl complexes were introduced via the alkyne bridge. The introduction of a chlorine atom at the aromatic ring reduced the activity against the COX-1 enzyme and increased the activity against COX-2. The 4-Cl chlorine complex, **144,** showed prominent activity against the HT-29 (IC_50_= 1.51 ± 0.12 µM) and MDA-MB-231 (IC_50_= 5.24 ± 0.33 µM) cancer cell lines, in comparison to cisplatin (IC_50_= 3.52 ± 0.48 µM- 3.57 ± 1.06 µM). Compound **144** also showed superior inhibition of COX-2 (69.9 ± 2.7%), compared to COX-1 (22.9 ± 1.7%). These complexes showed selectivity for cancer cells and were inactive against the non-cancerous cell line. These derivatives showed the process of apoptosis and demonstrated high penetration in the HT-29 cancer cell line, but there was less uptake in the MCF-7 and MDA-MB-231 cancer cell lines [120]. One year later, the same research group prepared cobalt-acetyl salicylic acid complexes, in which the fluorine atom was introduced at positions 3, 4, 5, and 6 in the aromatic ring to produce different derivatives. Compound **145** exhibited activity against the HT-29 and MDA-MB-231 cell lines, presenting IC_50_ values of 1.73 µM and 5.33 µM, respectively, upon evaluation via a crystal violet assay. Although these derivatives showed the inhibition of COX-2, they were nonselective for this enzyme [121]. The cobalt-alkyne acetylsalicylic acid complex served as the lead compound for further modifications and methyl-substituted derivatives were synthesized. These derivatives expressed better inhibition of COX-2, compared to COX-1. Compound **146** emerged as the most active anticancer agent against the HT-29 (IC_50_ = 1.25 ± 0.14 µM) cell line and also showed noticeable COX-2 (58.8 ± 9.8%) inhibition [122]. Zeise salt is a potassium trichlorido[ethylene]palatinate organometallic complex and has shown the capacity to inhibit the COX enzyme [123]. Weninger et al. synthesized the Zeise salt derivatives of acetylsalicylic acid, in which the two units were linked together using the alkyl linker groups. These derivatives (**147 a**–**d**) showed mild anticancer activity (IC_50_ = 30.1 µM–>50 µM) against the MCF-7 and HT-29 cancer cell lines. However, these derivatives presented the dominant inhibition of COX-1 and COX-2, selectivity was not achieved [124].

### 2.16. Stilbene Derivatives

Regulski et al. reported new trans-stilbene derivatives, along with their anticancer activity. Upon evaluation by an MTT assay, **148** (Figure 29) displayed prominent activity against the MCF-7 cell line. Derivative **148** also showed excellent inhibition of the COX-2 enzyme (IC_50_ = 0.09 µM; SI = 62.81), compared to the standard drug, celecoxib (IC_50_ = 0.11 µM; SI = 132.97). The presence of the nitro group was vital for the anticancer activity and the replacement of the nitro group by a chlorine atom produced less active derivatives. Similarly, the modification of one or more hydroxyl groups into methoxy groups produced active compounds. However, the exchange of the hydroxyl group with the methoxy group decreased the selectivity for COX-2 enzymes. The nitro group interacted with the Met522 of COX-2 by means of a sulfur-oxygen interaction and the hydroxyl group exhibited hydrogen bonding with Tyr355. Hydrophobic and pi-cationic interactions were also observed [125].

### 2.17. Hydrazone Derivatives

Senkarde and co-workers synthesized sulfonyl hydrazone derivatives as anticancer agents. Compound **149** showed prominent activity against the PC-3 (IC_50_ = 1.38 µM; SI = 432.30) cancer line, in comparison to cisplatin (IC_50_ = 3.3 µM, 22.2 µM) and showed selectivity for this cell line. The analog **149** exhibited excellent inhibition of COX-2 (91%) as compared to the COX-1 enzyme and showed the process of apoptosis. This compound carries a 2-chloro-3-methoxy phenyl group attached to the hydrazone linkage. In the molecular mode of action, the sulfonyl group of **149** presented hydrogen bonding with Tyr341 and Ser339 of COX-2. The hydrazine group showed hydrogen bond interaction with Val509 and the methoxy group with Ser516. Derivative **149** showed binding energy of −8.6 kcal/mol as compared to celecoxib (−12 kcal/mol). These derivatives followed the Lipinski rule of five and showed favorable physicochemical properties, such as molecular weight, logP, TPSA, logS, and molar refractivity values [126]. Popiolik et al. synthesized hydrazone-hydrazide derivatives of 5-bromo-2-iodobenzoic acid. Upon evaluation by MTT assay, analogs **150** (IC_50_ = 11.94 µM), **151** (IC_50_ = 22.20 µM) and **152** (IC_50_ = 4.11 µM) showed prominent anticancer activity against renal cell adenocarcinoma (769-P) and HepG2 cancer cell lines. The chlorine and nitro-substituted agents showed greater activity. Compound **151** is the nitro-substituted agent and showed excellent activity against the HepG2 cancer cell line. These derivatives showed low toxicity for the normal cell line. The derivative **153** presented slight inhibition of COX-2 (9.78%) and carries an indole ring attached through the aromatic linker that is important for its activity [127].

### 2.18. Peptides as COX-2 Inhibitors

Peptides have been used as therapeutic agents since 1920 when insulin was isolated from animal sources. With the passage of time, the synthesis of peptides became easier and various synthetic peptides were used as therapeutic agents. Nowadays, various medicinal chemistry techniques, such as combinatorial chemistry and high throughput screening, are being used to produce synthetic peptides with different pharmacological properties [128,129]. Ahmaditaba and co-workers used the solid-phase synthesis technique to produce peptide molecules. The COX-2 pharmacophores, such as 4-methylsulfonyl phenyl or 4-azidophenyl, were introduced (Figure 30). Compounds **155** (IC_50_ = 19.3 ± 0.02 µM, 19.9 ± 0.03 µM) and **156** (IC_50_ = 4.8 ± 0.12 µM, 5.7 ± 0.04 µM) exhibited anticancer activity against the HepG2 and A549 cancer cell lines, in comparison to the standard drug, celecoxib (IC_50_ = 16 ± 0.02 µM). Derivative **154** was the most active COX-2 inhibitor (IC_50_ = 0.2 µM, SI > 500) and demonstrated a prominent selectivity index, compared to the standard drug, celecoxib (IC_50_ = 0.06 µM; SI = 405). Analogs with the proline and tyrosine amino acids presented greater antiproliferative activities. Derivatives of the methylsulfonyl group showed better COX-2 inhibitory activities. Compound **154** showed two hydrogen bonds with the His90 and Arg513 of COX-2. The acidic functional group showed one hydrogen bond with Trp387. The phenyl group communicates with a hydrophobic pocket, formed by Leu531, Leu359, and Val349. Compounds **154** and **155** showed a good correlation between anticancer (IC_50_= 16 ± 0.13 µM and 12.1 ± 0.23 µM) and COX-2 (IC_50_ = 0.2 µM and 0.12 µM) inhibitory activities [130].

## 3. Conclusions

Cancer is a global health problem and the discovery of novel, effective, and safer agents is still of great importance. Herein, we have mainly focused on the anticancer compounds that have also shown COX-2 inhibition. Diarylheterocyclic compounds with pyrazole or pyrazoline as the main core are widely recognized COX-2 inhibitors. The pyrazole/pyrazoline scaffolds with sulfonylmethyl or sulfonamide groups have been attached to other molecules having anticancer properties, to produce new hybrid molecules. The majority of the molecules have been synthesized by the attachment of a pyrazole scaffold with other pharmacophores, such as coumarin, chrysin, caffeic acid, and other natural products. Hybrid molecules of pyrazole with coumarin have displayed excellent anticancer as well as COX-2 inhibitory properties. Hybrid molecules comprising caffeic acid and triazole have demonstrated prominent anticancer activity, accompanied by COX-2 inhibition. The addition of heterocyclic rings in the lead molecules increased the selectivity for COX-2 by increasing the intermolecular interaction with this enzyme. A correlation was also found between these two properties. The modification of NSAIDs such as naproxen has also produced highly potent anticancer molecules and few of them proved selective for COX-2. Similarly, the hybrid molecules of celecoxib and rofecoxib also proved highly effective. Many compounds exhibited anticancer activity by inducing apoptosis in cancer cells. The derivatives of heterocyclic compounds, such as cyanopyrimidine, pyrazolo-pyrimidines, pyrrolizine, and tetrazoles showed significant anticancer properties and better selectivity for the COX-2 enzyme. However, the presence of reactive groups in some compounds makes them unfavorable for in vivo assays and drug development. This review highlights the recent heterocyclics and hybrid anticancer molecules that have displayed anti-inflammatory and COX-2 inhibitory properties.

## Figures and Tables

**Figure 1 pharmaceuticals-15-01471-f001:**
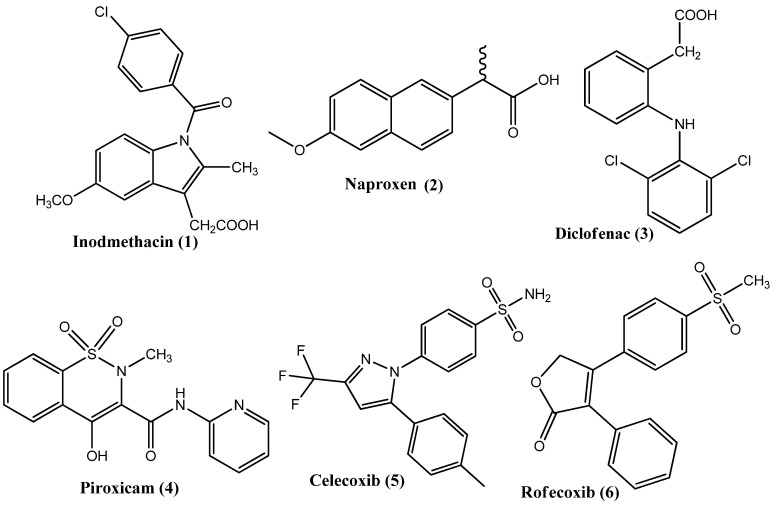
Structural formulas of non-selective and selective NSAIDs.

**Figure 2 pharmaceuticals-15-01471-f002:**
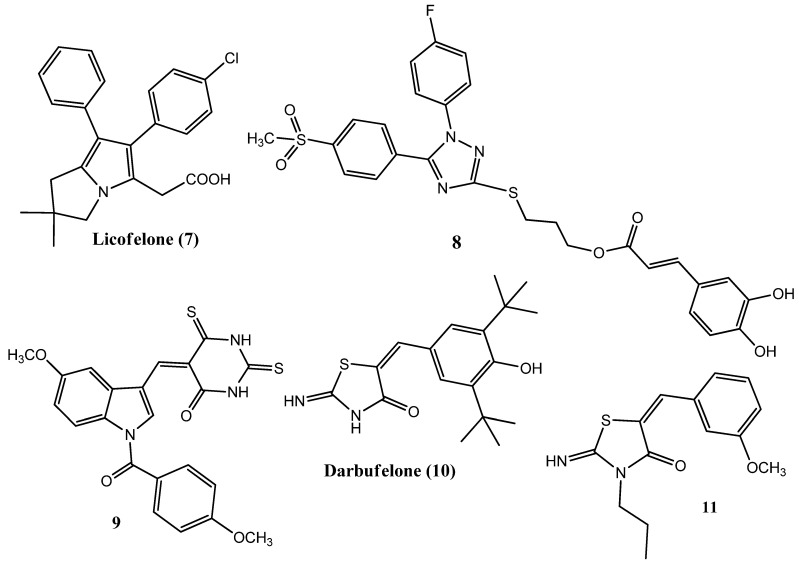
Structure of the lead compounds with anticancer and COX-2 inhibitory potential.

**Figure 3 pharmaceuticals-15-01471-f003:**
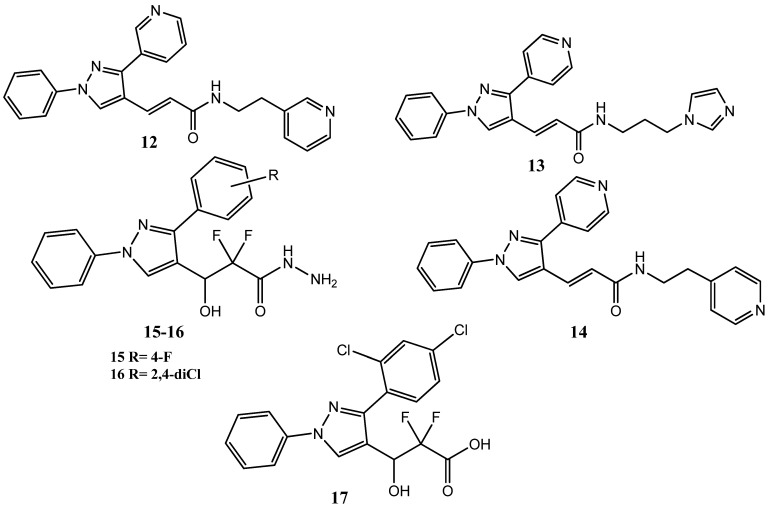
The 1,3–diaryl pyrazole derivatives.

**Figure 4 pharmaceuticals-15-01471-f004:**
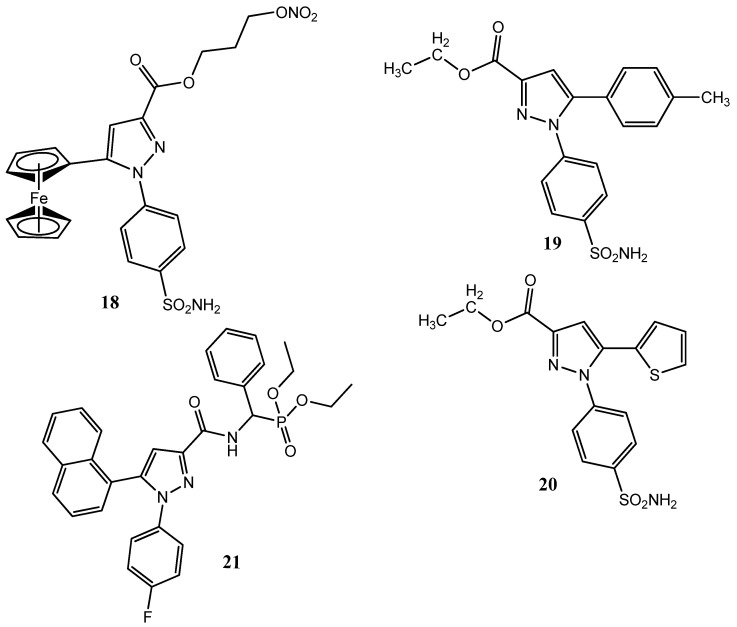
Pyrazole derivatives with ferrocene (**18**), ester (**19**, **20**), and aminophosphonyl (**21**) groups.

**Figure 5 pharmaceuticals-15-01471-f005:**
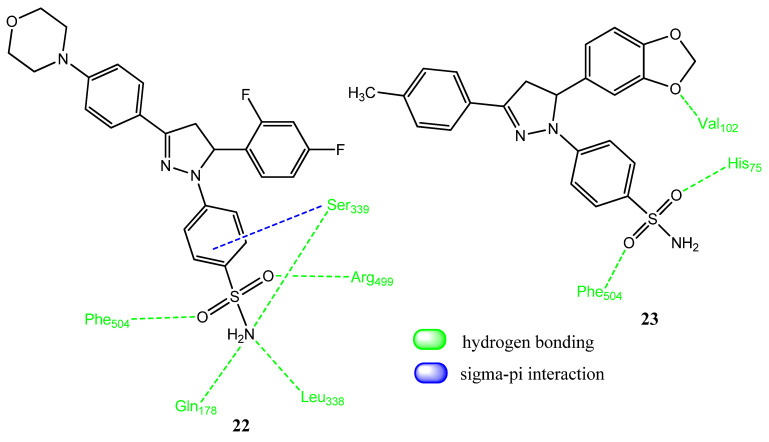
Pyrazoline derivatives and their docking interactions with COX-2.

**Figure 6 pharmaceuticals-15-01471-f006:**
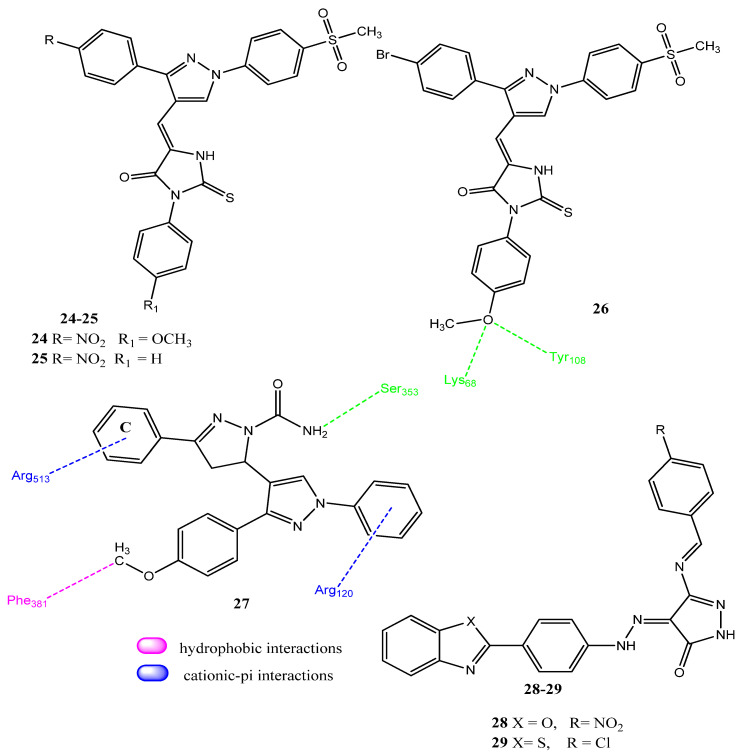
Pyrazole–based hybrid molecules and their interactions with COX-2.

**Figure 7 pharmaceuticals-15-01471-f007:**
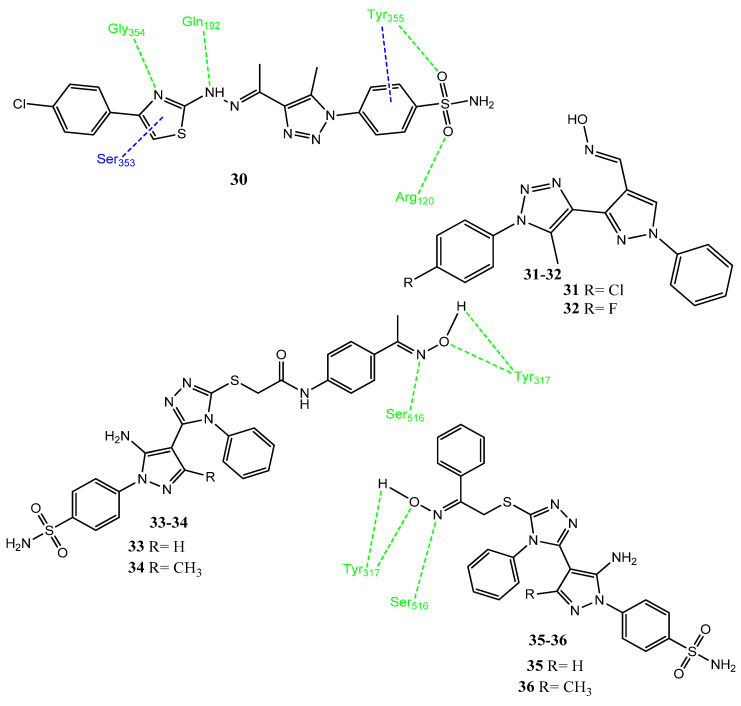
Pyrazole–based hybrid molecules.

**Figure 8 pharmaceuticals-15-01471-f008:**
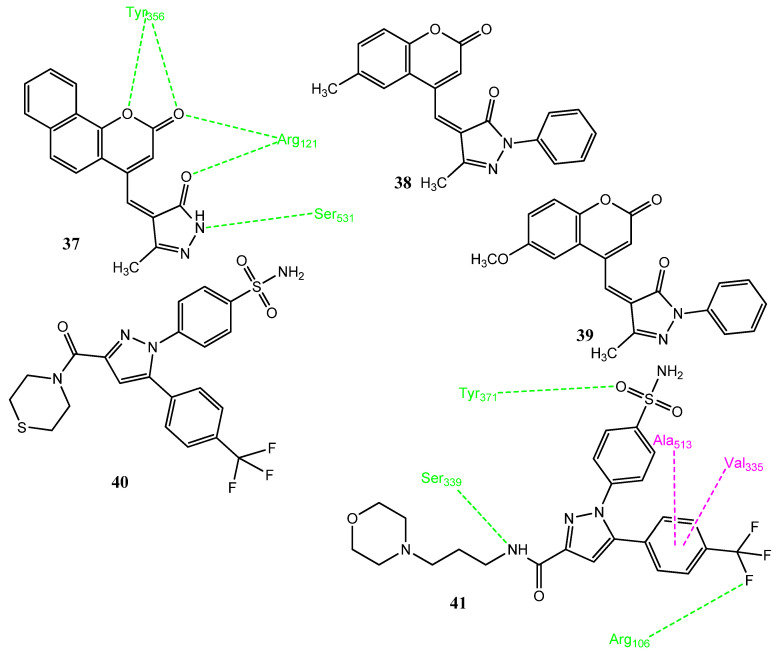
Pyrazole and pyrazolone–based hybrid molecules.

**Figure 9 pharmaceuticals-15-01471-f009:**
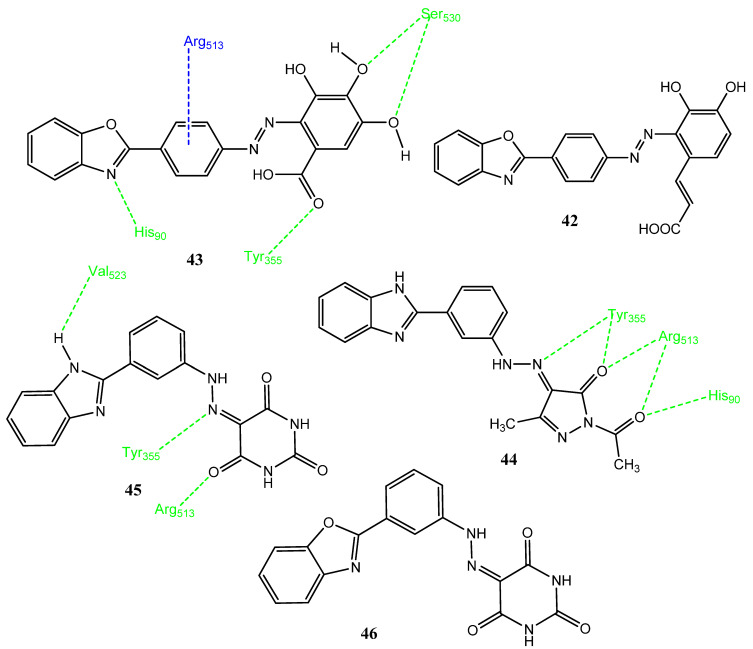
Benzimidazole and benzoxazole hybrid molecules and their interactions with COX-2.

**Figure 10 pharmaceuticals-15-01471-f010:**
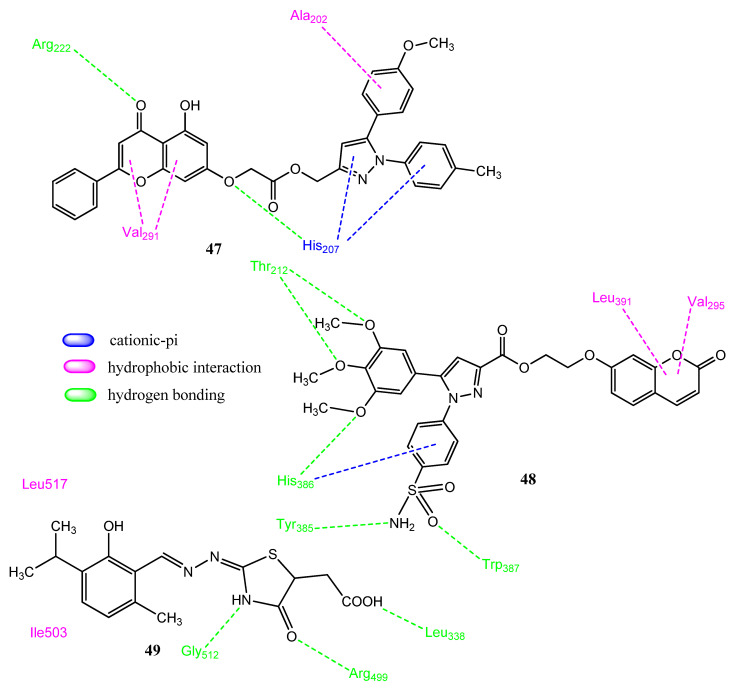
Structures of natural product–based hybrid molecules and their interactions with COX-2.

**Figure 11 pharmaceuticals-15-01471-f011:**
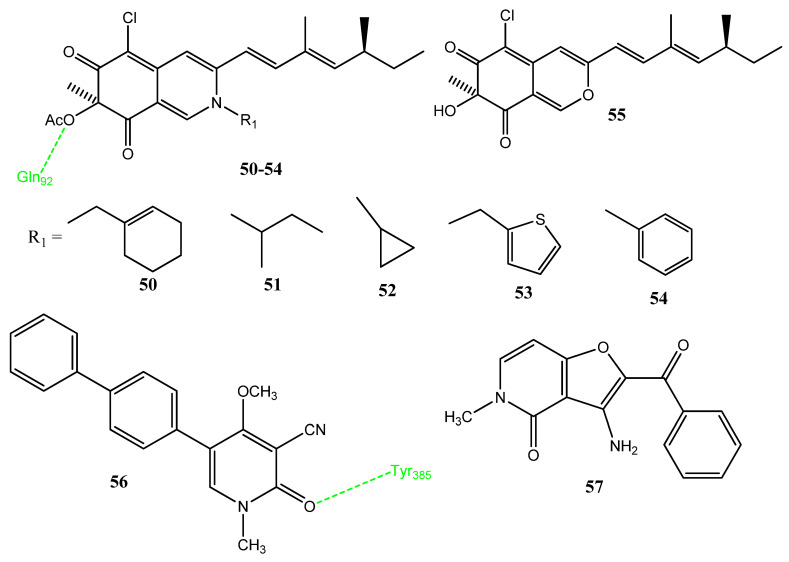
Derivatives of sclerotiorin and ricinin.

**Figure 12 pharmaceuticals-15-01471-f012:**
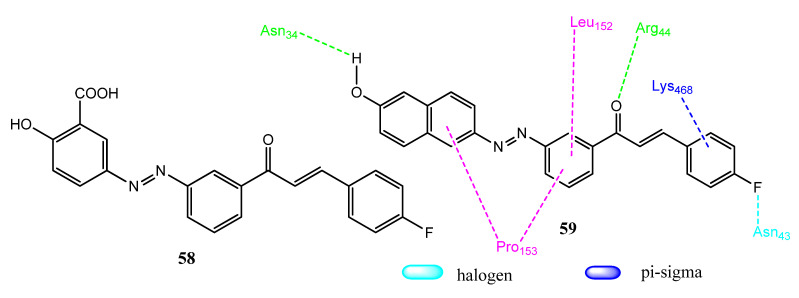
Chalcone–derived compounds.

**Figure 13 pharmaceuticals-15-01471-f013:**
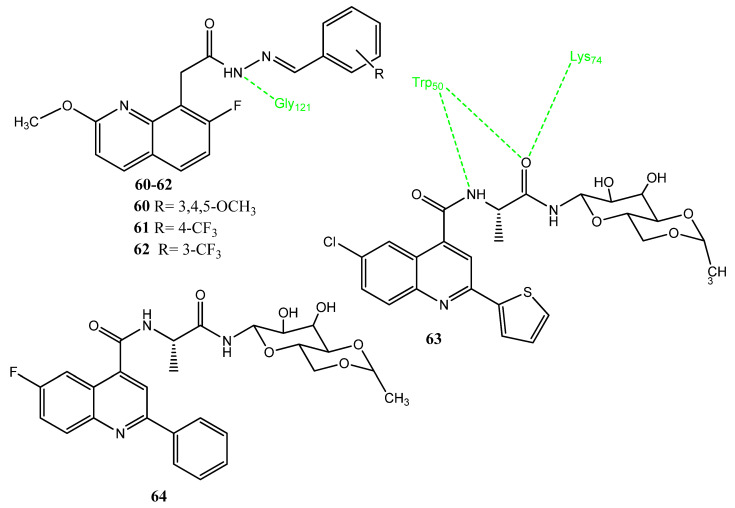
Quinoline–based molecules.

**Figure 14 pharmaceuticals-15-01471-f014:**
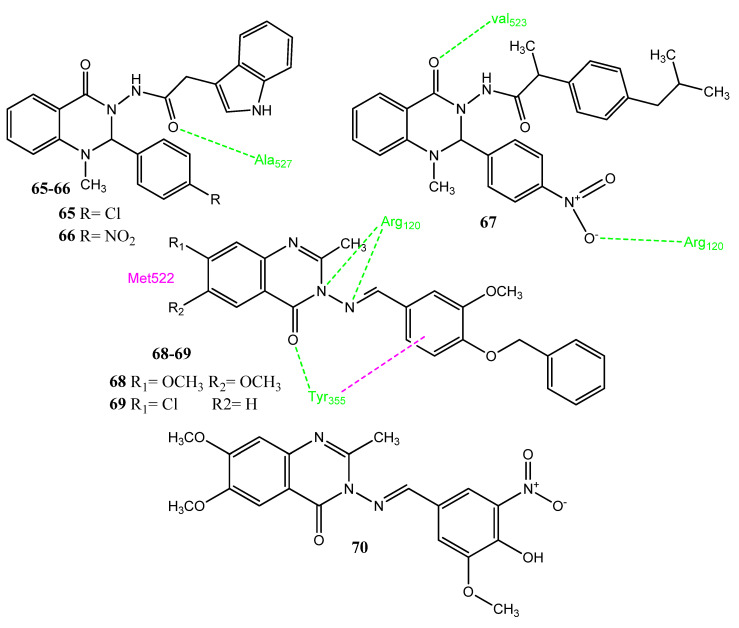
Quinazolinone derivatives.

**Figure 15 pharmaceuticals-15-01471-f015:**
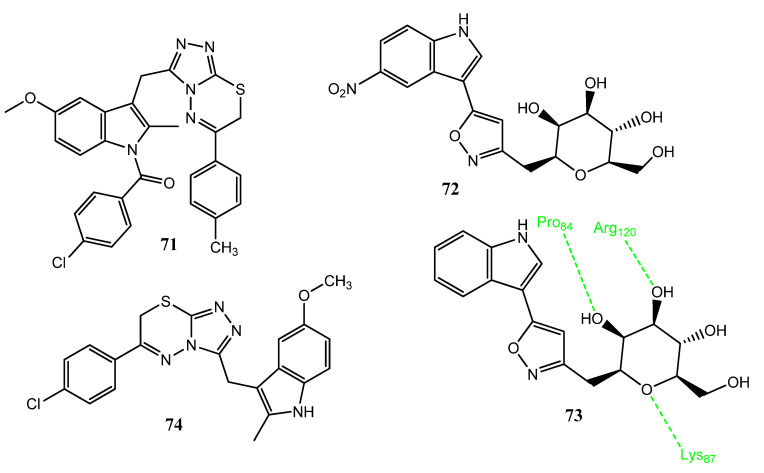
Indole derivatives.

**Figure 16 pharmaceuticals-15-01471-f016:**
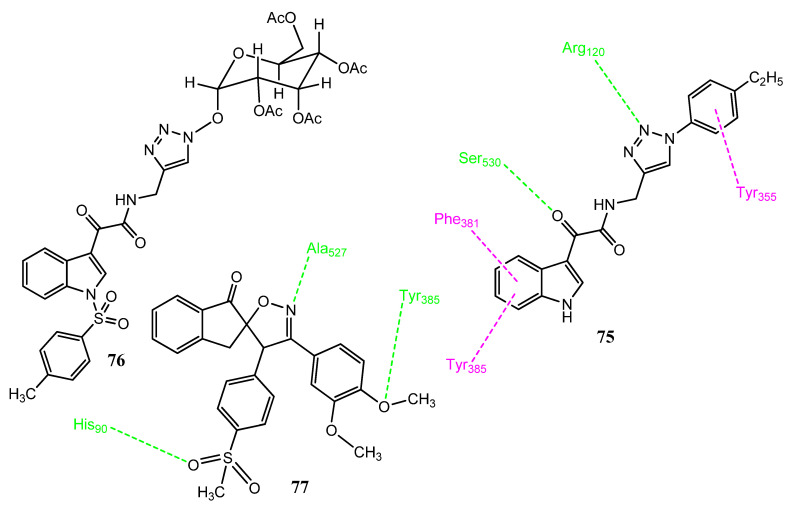
Indole- and indanone-based molecules.

**Figure 17 pharmaceuticals-15-01471-f017:**
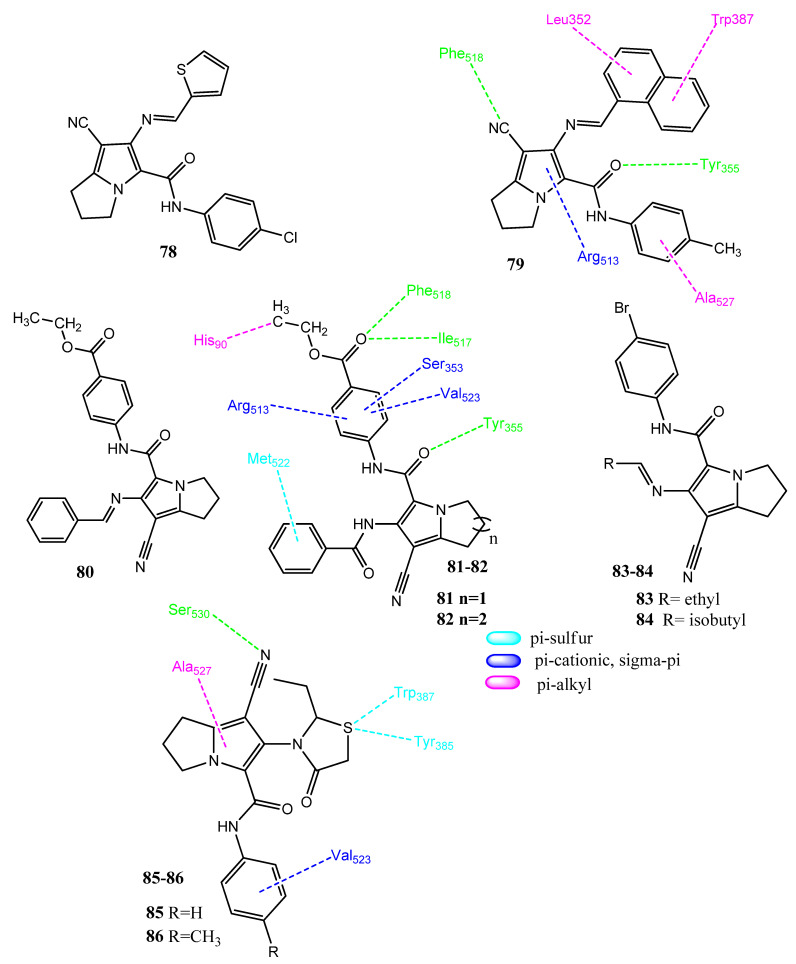
Pyrrolizine derivatives.

**Figure 18 pharmaceuticals-15-01471-f018:**
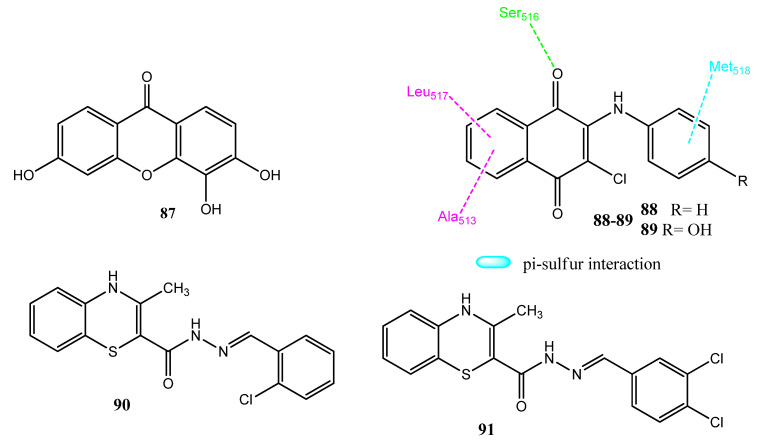
Xanthone, naphthoquinone, and benzothiazine derivatives.

**Figure 19 pharmaceuticals-15-01471-f019:**
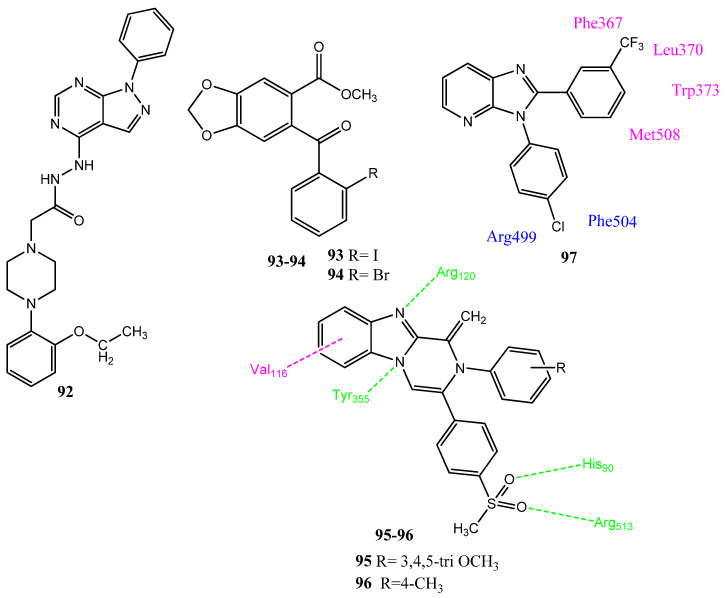
Xanthone, naphthoquinone, and benzothiazine derivatives.

**Figure 20 pharmaceuticals-15-01471-f020:**
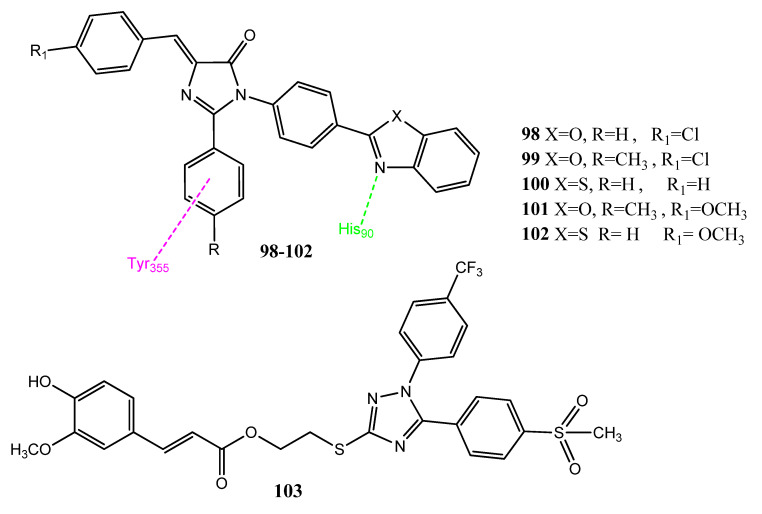
Imidazolone and triazole derivatives.

**Figure 21 pharmaceuticals-15-01471-f021:**
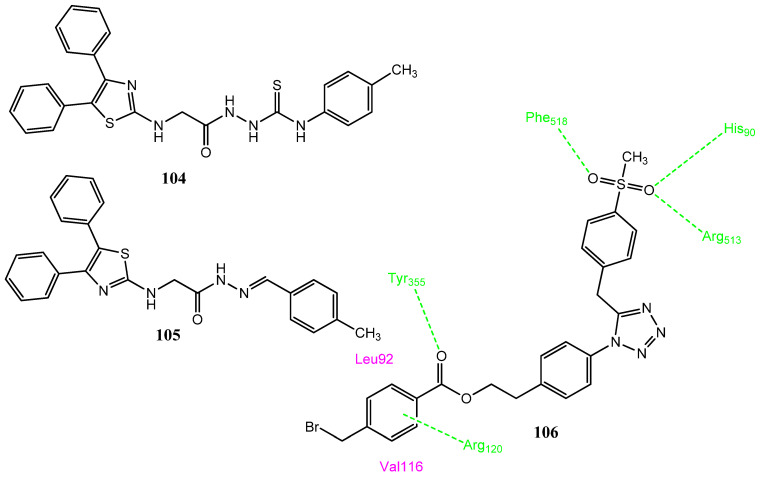
Thiazole and tetrazole derivatives.

**Figure 22 pharmaceuticals-15-01471-f022:**
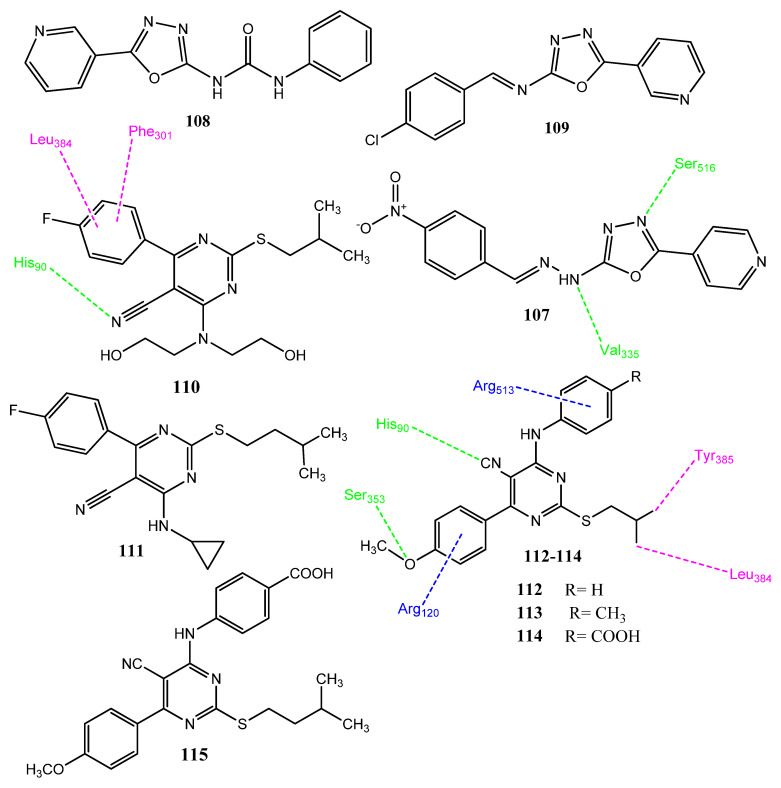
Oxadiazole and pyrimidine derivatives.

**Figure 23 pharmaceuticals-15-01471-f023:**
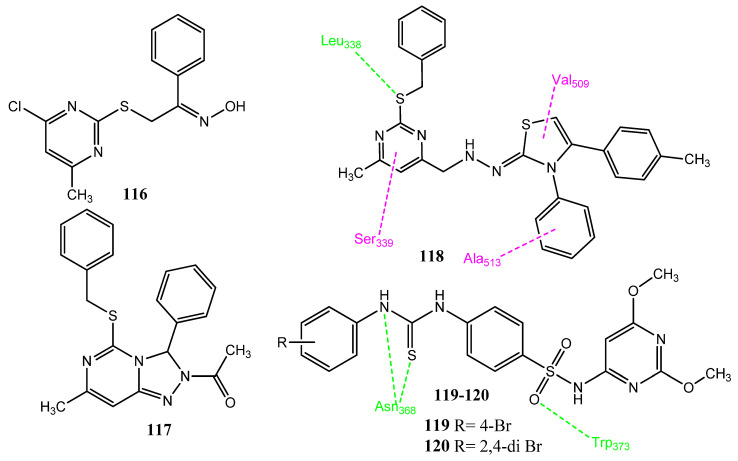
Heterocyclic compounds.

**Figure 24 pharmaceuticals-15-01471-f024:**
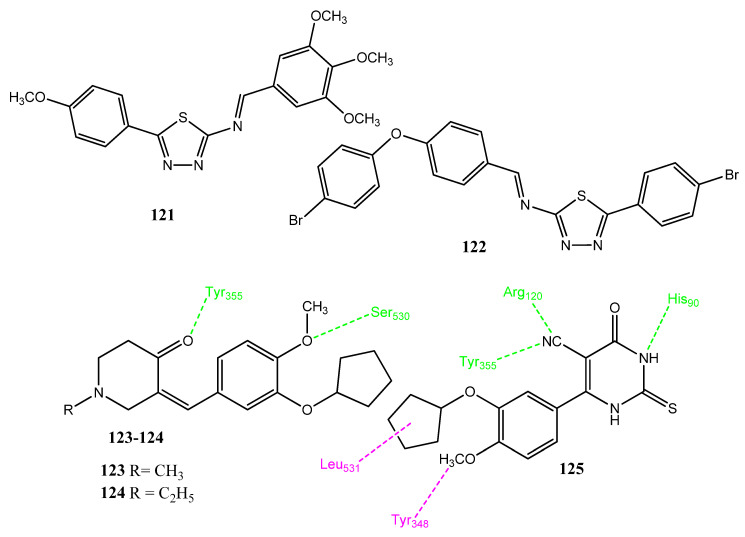
Thiadiazole and cyclopentolate derivatives.

**Figure 25 pharmaceuticals-15-01471-f025:**
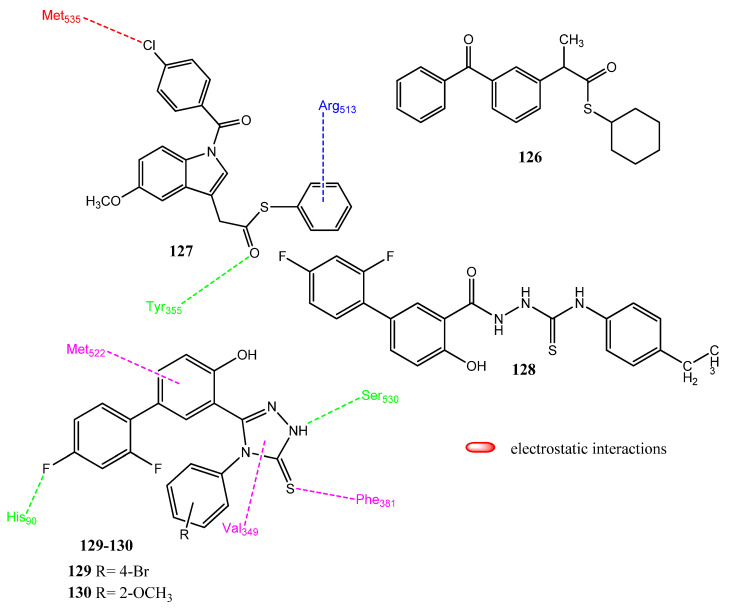
NSAIDs derivatives.

**Figure 26 pharmaceuticals-15-01471-f026:**
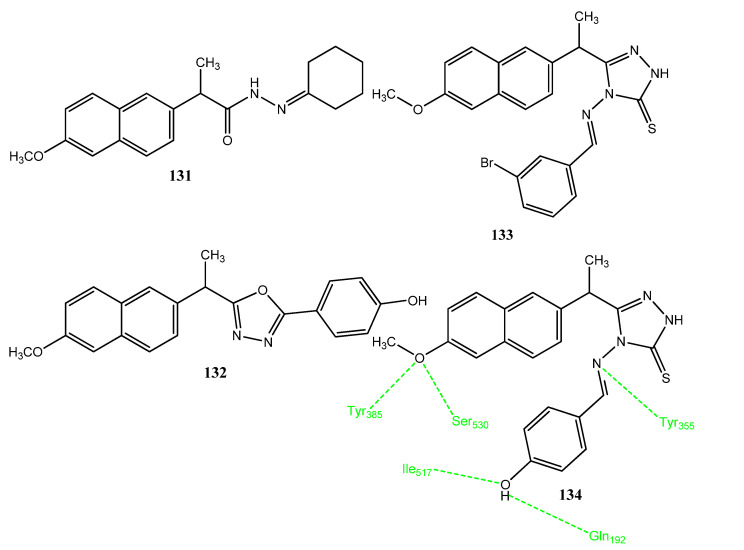
The derivatives of naproxen and their connection with COX-2.

**Figure 27 pharmaceuticals-15-01471-f027:**
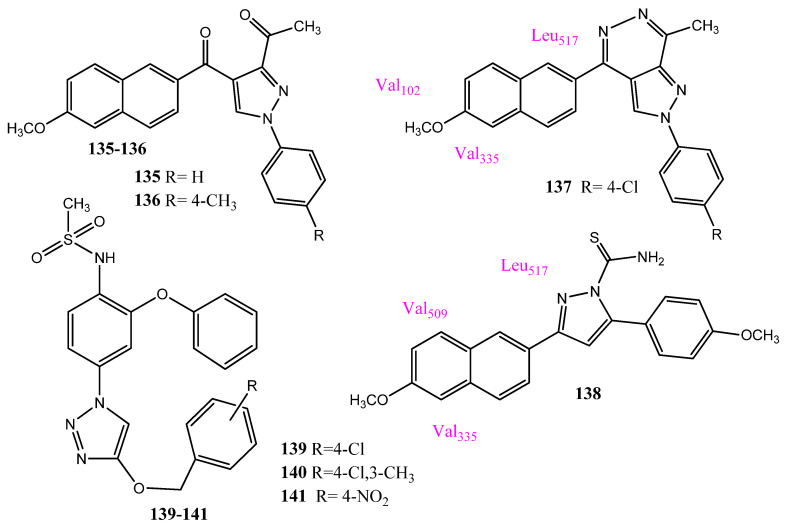
Naproxen and nimesulide derivatives.

**Figure 28 pharmaceuticals-15-01471-f028:**
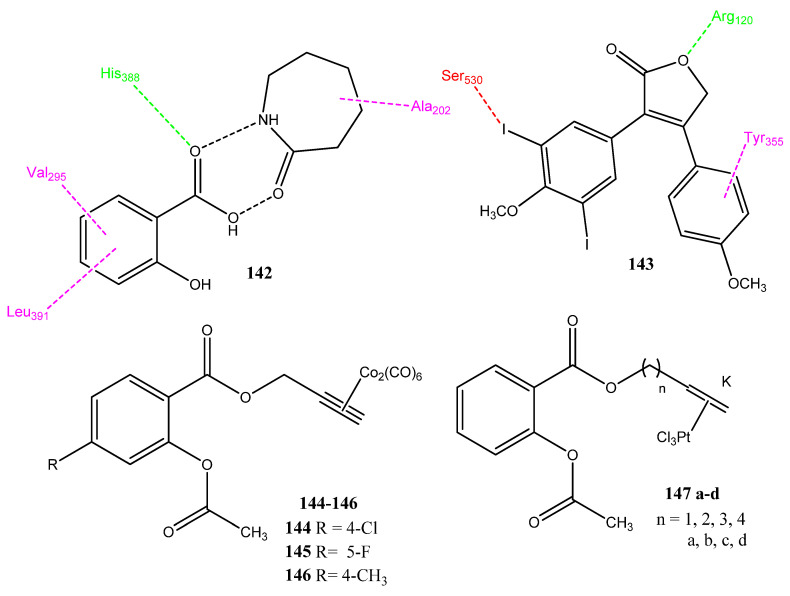
Salicylic acid, rofecoxib, and acetylsalicylic acid–based metal complexes.

**Figure 29 pharmaceuticals-15-01471-f029:**
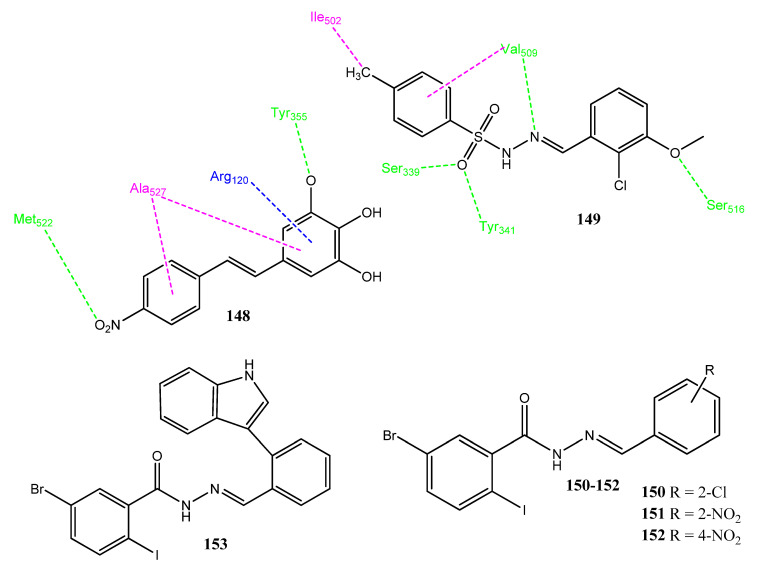
Stilbene and hydrazone derivatives.

**Figure 30 pharmaceuticals-15-01471-f030:**
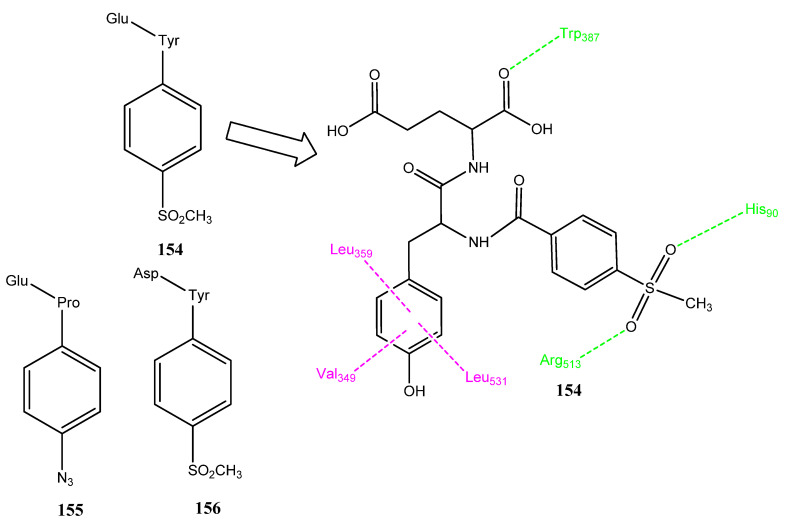
Peptides as COX-2 inhibitors.

## Data Availability

Data sharing not applicable.

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
