# Peer review of "Cyclooxygenase-2 (COX-2) as a Target of Anticancer Agents: A Review of Novel Synthesized Scaffolds Having Anticancer and COX-2 Inhibitory Potentialities"

_pharmaceuticals, 2022, doi:10.3390/ph15121471_

Round 1
Reviewer 1 Report
Authors provide meticulous description of various classes of organic compounds synthesized in order to achieve both anticancer and COX-2 inhibitory activities.
Review will benefit from including the following information:
1. How much of anticancer activity of synthesized compounds is due to its ability to inhibit COX-2 and how much is independent of this inhibition? Does their anticancer activity is present in cancers (and cells) which do not express COX-2?
2. Are there any clinical trials conducted with described compounds? At what stage they are? Are the results promising?
3. Is it possible to include a table which will compare anticancer and COX-2 inhibitory activities of all (or some) of described compounds?
Moderate English changes are required. Just couple of examples of sentences which require editing are shown below.
Lane 55 – In the current wording the sentence states that inhibition of COX-2 induces metastasis. I guess, it is not what authors mean. The sentence must be modified for clarity.
Lane 73 – what means “compound 8 also expressed the process of apoptosis”? Do authors mean that it induces apoptosis?
Reviewer 2 Report
The authors Noor ul Amin Mohsin and co-workers submitted their manuscript entitled “Cyclooxygenase-2 (COX-2) as a target of anticancer agents: A review of novel synthesized scaffolds having anticancer and COX-2 inhibitory potentialities” to the journal “Pharmaceutical” in order to be considered for publication as a “Review”.
The review addresses a very important topic, namely the anticancer activity of compounds by inhibition of COX-2. In their update, the authors refer to recent work from 2017-2022. The compounds are presented sorted by their chemical structure or core, i.e. different heterocycles, derivatives of natural products, derivatives of NSAIDs, hybrid molecules. The authors pick up the biological effects, i.e. mainly cytotoxicity towards cancer cell lines and inhibition of COX-2. Another focus is on which molecule parts interact with amino acids of the isoenzyme COX-2.
Overall, the authors provide a very comprehensive overview. The manuscript seems suitable in principle to be published in the Special Issue. Nevertheless, there are several places where literature should be added to make the presentation even more complete. In addition, the manuscript can still be formally optimized. However, the English seems to be okay with minimal exceptions.
I went to the effort to help the authors further strengthen their manuscript and facilitate publication in “Pharmaceuticals." Please find my particular comments below:
Reference [1] is more than 10 years old can be updated, considering the works of Siegel et al. from 2022, e.g., https://doi.org/10.3322/caac.21708, https://doi.org/10.3322/caac.21754, https://doi.org/10.3322/caac.21731, https://doi.org/10.3322/caac.21718.
Mainly, it is about the two isoforms of COX, i.e. COX-1 and COX-2. However, recent research also discusses the so-called COX-3. The authors are kindly asked at least to mention it in their manuscript, e.g. https://doi.org/10.1073/pnas.16246869, 10.1073/pnas.222543099.
@ chemical structures: Please standardize the representation in the structures, e.g. of the methyl groups or how alkyl/methylene units are represented. The Cl-substituents in structure 17 look somewhat strange (also among other structures with Cl-substituent).
Structure 9: Please define R and Ar in the chemical structure.
Figure 2: Do the compounds presented in Figure 2 only exhibit inhibition of COX-2 or also of COX-1?
The subdivision into parts 2. and 3. makes little sense, because both consider compounds with COX-2-inhibitory and anticancer properties. Under heading 3, the compounds are only sorted according to their chemical scaffold. The authors are kindly asked to reconsider this.
“compounds having pyrazole scaffold have drawn major attention as anticancer agents due to their anti-inflammatory and anticancer activities”. The authors are asked to consider work published on that topic, such as 10.1016/j.bioorg.2019.103470, 10.2174/13895575113139990078, 10.3390/molecules26113439.
“The presence of the aminosulfonyl or sulfonylmethyl group is also very significant for the COX-2 selectivity” Please add a short reason for this in your manuscript.
Structure 15-16: Maybe better connect “R” to the aryl ring in the manner like it was done in Structure 60-62.
Figure 4: The caption is not very specific, i.e. please comment which of the compounds is bearing ferrocene, aminophosphonyl, or ester groups.
@ pyrazole derivatives: The authors are kindly asked to consider recent work: https://doi.org/10.1016/j.ejmech.2019.03.052, https://doi.org/10.1515/znc-2021-0217.
“Pyrazoline is a reduced form of pyrazole and has achieved great utility in new drug synthesis.” It is suggested to consider publication on that topic, such as 10.2174/157489109789318569, https://doi.org/10.1016/j.ejmcr.2022.100042. Also 3.3 hybrids: 10.3390/ijms21155507.
“Molecular hybridization is a trend in drug design and discovery in which the pharmacophores of active drug molecules are combined to produce new molecules.” It is suggested to cite literature on that topic such as: 10.2174/1568026619666190619115735, 10.2174/092986707781058805.
“Derivatives 24, 25 and 26 demonstrated prominent anticancer potential against MCF-7, A549, and HCT-116 187 cell lines than doxorubicin”? Please check this sentence considering its meaning.
“The authors tried to boost the anticancer and anti-inflammatory activities” How was this approach performed in particular? Was is done using the nitric oxide-releasing group? Please clarify in the manuscript. AND: “These compounds also showed a prominent release of nitrous oxide (NO)” Please shortly mention the meaning of NO relase in the context of anticancer activity.
“The sulfamoyl-substituted derivatives showed greater activity as compared to sulfonylmethyl and un-substituted analogues” Is it possible to provide a reason for this behavior. If so, please add to the manuscript.
“distance of 2.33-3.11 °A” Please adjust the sign of Angström.
“The longer linker group between pyrazole and coumarin may be the reason of increased potency” Is it possible to provide a more detailed reason in the manuscript?
Structure 49: Interaction (dashed line) of Leu517 and Ile503 are missing. (Besides, take care about consistent use of font). The dashed line should also be added to other structures, e.g., 97, 137, 138
“Marine natural products showed excellent biological activities and structural modification of plant-derived compounds can enhance the pharmacological activities of the lead compound.” It is suggested to include references on that topic, e.g., https://doi.org/10.1080/13543776.2022.2012150, https://doi.org/10.1016/j.biotechadv.2021.107871, https://doi.org/10.4155/fmc.11.118.
“the Lipinski rule of five except one violation i.e. logP value not greater than 3”. According to the Rule of Five the logP value should not exceed 5(!). By the way, the abbreviation RO5 is introduced later in the manuscript. Please revise to explain the abbreviation at the first occurrence in the manuscript and then use it throughout the manuscript. Take care, sometimes it is Lipinski or Lipinski’s rule. Please be consistent with that.
“Structure 68 showed favourable pharmacokinetic properties…” What does this mean in particular?
“In molecular docking investigations, 74 exhibited interactions with the COX-2 active site”. Which amino acids in particular?
“and 76 (IC50 = 18.53) showed” insert unit “μM”… and μmol/L should be μM.
Structure 81-82: “n” is missing in the formula at the brackets.
The IC50 values given in μg/ml should be addressed as μM to allow for an easier comparison. Maybe it is also possible to be consistent with IC50. In some cases it is termed as GI50.
“NSAIDs have also received the attention of medicinal chemists to produce new anticancer agents.” Please provide references on that, e.g., https://doi.org/10.1093/jnci/94.4.252, https://doi.org/10.2147/CMAR.S175212, https://doi.org/10.1515/znc-2020-0093, https://doi.org/10.1186/s13058-020-01343-1.
“The structure of the new molecule preserved the cis configuration of combretastatin and prevented its conversion into trans form”. Please explicitly mention the advantage of this circumstance.
“Obermoser and co-workers synthesized new chlorine-substituted acetylsalicylic acid derivatives in which Cobalt carbonyl complexes” The group also investigated the F and CH3 analogues (https://doi.org/10.1039/C9DT03330K, https://doi.org/10.1002/ardp.202100408) and platinum molecules https://doi.org/10.3390/ijms19061612. Please add to the manuscript.
“Compound 146 showed prominent activity against the PC-3 (IC50 = 1.38 μM; SI = 432.30) cancer line than cisplatin” more prominent… than cisplatin.
“properties like logP, TPSA, logS values etc” What means “etc”. That seems to be more of a filler word.
“Peptides as COX-2 inhibitors” It is suggested to introduce one or two sentences at the beginning of the paragraph highlighting the meaning / advantages of peptides in drug discovery, considering recent literature such as https://doi.org/10.1038/s41573-020-00135-8, https://doi.org/10.1038/s41392-022-00904-4, https://doi.org/10.1021/acs.jmedchem.7b00318, https://doi.org/10.1016/j.bmc.2017.06.052.
“inducing the phenomenon of apoptosis” This phrase occurs several times in the manuscript. It is highly suggested to term it “inducing apoptosis” – or is apoptosis really a phenomenon among cell death modalities? Rather not…
It is suggested to standardize the number of decimal places for the given values (e.g. inhibition of COX-2) throughout the manuscript. I think basically one decimal place is enough. Also with the IC50 cytotoxicity values, a specification to the third decimal place makes little sense. Moreover, when reporting a mean value, the Standard deviation/error cannot have more digital places i.e. the SD/SE would be more accurate than the mean value.
Another thing that gives me pause is related to inhibition of COX. On the one hand, in the introduction, when briefly introducing COX-1 (and COX-3), the authors should address why inhibition of COX-1 is undesirable and therefore selectivity for the isoenzyme COX-2 is sought. Perhaps, in the course of the manuscript, the authors will be able to give some examples of the structural modifications that make it possible to shift selectivity in favor of COX-2. In addition, I see the specification of the values for inhibition of the enzyme somewhat critically. This always depends on the concentration used. However, the authors hardly say anything about this. I think, however, that the data would be important.
Although there will be a close editing by the MDPI publisher, the authors are kindly asked to correct formal inconsistencies, such as: Bold (e.g. of compound numbers), use of space (e.g. ±), introduce abbreviations (e.g. explanation when used for the first time, e.g., DPPH, EFGR), consistency (e.g. position…, with or without #), caption (e.g. Ki value), different fonts in the Figures should be unified, punctuation, use of in italics (e.g. in vivo/vitro/silico; or elsewhere just not, i.e. Figure), use of dash (e.g. Phe-182 or Phe182 / anticancer vs. anti-cancer, in-vivo vs. in vivo / HT-29 vs HT29, ), use of Greek symbols (e.g. pi interaction), use of “&” is not scientific style, among others. This listing is not understood to be complete. Please thoroughly revise the manuscript.
Dear authors, I know that you now have a lot to work on once again. But I am confident that it will be worth the effort and that you will be able to enrich your manuscript once again. All the best!
Round 2
Reviewer 2 Report
The authors Noor ul Amin Mohsin and co-workers provided a revised version of their manuscript “Cyclooxygenase-2 (COX-2) as a target of anticancer agents: A review of novel synthesized scaffolds having anticancer and COX-2 inhibitory potentialities” submitted to the MDPI journal “Pharmaceuticals” in order to be considered for potential publication as a “Review”.
The authors acted on every concern or suggestion (e.g., including particular explanations or slight modifications to make things even clearer) or provided rebuttal where it is due. Especially, much effort was made in adding recent studies to their manuscript to make their review both more comprehensive and up to date. Some minor formal inconsistencies do not necessarily worsen the study and will be corrected during the proof-reading.
All in all, further processing of the manuscript is recommended. All the best!